



# Explainable machine learning for modelling of net ecosystem exchange in boreal forest

Ekaterina Ezhova[1,*], Topi Laanti[2,*], Anna Lintunen[3], Pasi Kolari[1], Tuomo Nieminen[1], Ivan Mammarella[1], Keijo Heljanko[2,4], and Markku Kulmala[1]

[1]INAR/Physics, University of Helsinki
[2]Department of Computer Science, University of Helsinki
[3]INAR/Agricultural and Forest Sciences, University of Helsinki
[4]Helsinki Institute for Information Techonology HIIT
[*]These authors contributed equally to this work.

**Correspondence:** Ekaterina Ezhova, ekaterina.ezhova@helsinki.fi

**Abstract.** There is a growing interest in applying machine learning methods to predict net ecosystem exchange (NEE) based on site information and climatic variables. In case of successful performance, it could give an excellent opportunity for gapfilling or upscaling, i.e., extrapolation of results to times and sites for which direct measurements are unavailable. There exists already quite an extensive body of research covering different seasons, time scales, number of sites, input variables (features), and

models. We apply four machine learning models to predict the NEE of boreal forest ecosystems based on climatic and site variables. We use data sets from two stations in the Finnish boreal forest and model NEE during the peak growing season and the whole year. Using Explainable Artificial Intelligence methods, we compare the most important input variables chosen by the models. In addition, we analyze the dependencies of NEE on input variables against the existing theoretical understanding of NEE drivers. We show that even though the statistical scores of some models can be very good, the results should be treated

with caution, especially when applied to upscaling. In the model setup with several interdependent variables ubiquitous in atmospheric measurements, some models display strong opposite dependencies on these variables. This behavior might have adverse consequences if models are applied to the data sets in future climate conditions. Our results highlight the importance of Explainable Artificial Intelligence methods for interpreting outcomes from machine learning models, particularly when a large set of interdependent variables is used as a model input.

## 1 Introduction

Forests play an important role in the global carbon cycle because they remove carbon from the atmosphere through photosynthesis and store it in the wood biomass and forest soil. Recent studies suggest that in the past several decades, the net carbon uptake of the boreal forest has been increasing and that of the tropical forest - decreasing, making the boreal forest the largest terrestrial carbon sink on the planet (Tagesson et al., 2020). Although the dynamics of the forest carbon cycle and its drivers

are, in general, well-understood, the interaction of forests with various climatic drivers, their response to climate change, and at the same time ability to mitigate it, continue keeping boreal forests in a focus of multidisciplinary research at all levels



from observations to global modeling (Artaxo et al., 2022; Petäjä et al., 2022; Kulmala et al., 2020, 2023; Tang et al., 2023). There is a growing need for more accurate models of carbon fluxes, providing reliable results in warming climate conditions (Kämäräinen et al., 2023). Hence, suitable models must correctly capture current carbon cycle dynamics using commonly mea-

sured ecosystem-level data and give reasonable predictions for, e.g., future higher temperatures. In other words, the models' performance should be adequate in the range of variable values currently underrepresented in the data sets used for model training.

The net carbon efflux of an ecosystem, accounting for both uptake of $CO_2$ in the process of photosynthesis and release of $CO_2$ due to respiration, is typically measured using the micrometeorological eddy covariance method (Hicks and Baldocchi,

2020). Several meteorological and environmental factors such as solar radiation, air temperature, and humidity have an impact on the Net Ecosystem Exchange (NEE), and they are widely used as input variables in process-based models (Launiainen et al., 2022; Junttila et al., 2023). The importance of these factors for NEE varies with the season. Currently, there is plenty of NEE data available from the FLUXNET database, as well as extensive meteorological reanalysis data sets or measurements of many different variables directly from research stations. Data availability boosted the application of data-intensive machine learning

(ML) methods to NEE modeling (Dou and Yang, 2018; Zeng et al., 2020). Many different ML models have been tested, but Random forest has appeared particularly popular and suitable for this task (Liu et al., 2021; Reitz et al., 2021).

ML models play an important role in providing an alternative for the hypothetic-deductive modeling approach, i.e., an inductive approach. This means no prior assumptions are made about the data, which is modeled with a purely empirical model with a general function class. Using ML, the functional relationship between carbon flux (net ecosystem exchange,

gross primary production or respiration) and the site and climatic variables, including radiation, meteorological and biospheric input, can be obtained. However, these empirical machine learning models are often "black box" in the sense that the weights that the models use to make the predictions can not be directly extracted from the model to provide a human understandable way to interpret them easily. The results, therefore, should be treated cautiously. Recently, Shirley et al. (2023) demonstrated with an example from Alaska that the boosted regression tree ML model gave inaccurate results in current and future carbon

balance estimates at high latitudes. Increasing the data set by adding more stations from the same area improved the result for the current carbon sink. Still, future estimates were unreliable, ascribed to the fact that the data sets representing future conditions were missing from the data sets used for model training.

There exists plenty of literature featuring the ML approach to quantify different components of the carbon cycle using site and climatic variables as input (Dou and Yang, 2018). In general, most of the studies consider daily or annual fluxes, more

seldom subhourly. This depends much on the focus of the work: upscaling studies that seek to provide spatial extrapolation of existing measurements to larger regions operate more often on daily and annual time scales. Gapfilling studies focusing on filling in data missing in time series often use a subhourly scale. Finally, 'find the best model'-studies typically also concentrate on a daily scale. Upscaling studies deal both with regional (Zhu et al., 2023; Shirley et al., 2023) and global (Zeng et al., 2020) spatial scales. In these studies, input variables are typically partly measured or adopted from reanalyses (climatic variables,

such as temperature, radiation, etc.) and partly derived from the satellite observations (biological variables characterizing the ecosystem, e.g., leaf area index). In studies that aim to find the best model, a more extensive set of input variables is typically




used, including humidity-related variables and possibly different turbulent fluxes, i.e., components of surface energy balance (Cai et al., 2020; Wood, 2021). In all outcomes reported so far, an increase in the number of input variables always improved the model performance, even though this improvement could be quite modest.

The development of ML approaches to carbon cycle analysis has been progressing in parallel, with some studies focusing on specific sites or clusters of vegetation with the same plant functional type (e.g., Liu et al., 2021; Wood, 2021). In contrast, others focus on global or regional upscaling (e.g., Zhu et al., 2023; Zeng et al., 2020). Earlier studies did not emphasize the details of the predictive model, but more recently, there has been a growing emphasis on understanding the importance of different input variables in these models (Kämäräinen et al., 2023). However, with many high-performing models functioning as "black

boxes," there is a rising need for tools to make ML models' operations and decision-making processes more interpretable.

In response to this need, various methods that attempt to make ML models more open and interpretable have emerged, allowing researchers to gain insights into the factors influencing the models' predictions. The methods that mitigate these problems and help to clarify the effect of the input variables (features) on the performed predictions in machine learning models are called explainable artificial intelligence (XAI) methods (Dwivedi et al., 2023). They aim to reduce uncertainty,

providing more confidence in the results and ensuring that the models consistently capture the relationship between input and predicted variables. This is crucial when the models are used to make future predictions. With XAI techniques, researchers can explore and analyze the factors that influence the outcomes, making it easier to interpret the results and enhance the utility of ML approaches in the context of carbon cycle research.

In the present study, we model boreal forest NEE with subhourly time resolution, using an extensive set of input variables

from two research stations at different latitudes: Hyytiälä at 61°51'N and Värriö at 67°46'N. Using the same time resolution, we use different data sets considering separately the peak growing season (defined as the period of maximum photosynthetic activity of an ecosystem) and the whole year. One of the data sets is divided into pre- and post-thinning periods due to the thinning of forests significantly impacting different measurements regarding many site variables.

We expect an ML model to learn differently depending on both the time resolution and seasonality of the time series used

for model training. For example, diffuse radiation is an essential input variable for photosynthesis on a subhourly scale during the peak growing season because ecosystem photosynthesis is enhanced under higher diffuse radiation conditions due to better light use efficiency (Gu et al., 2002; Ezhova et al., 2018). In winter, this effect is missing, which might make diffuse radiation not as crucial variable for the model trained on the whole year data set. Instead, other input variables, such as air or soil temperature, can be relevant when focusing on the seasonal cycle of carbon fluxes (Kolari et al., 2009). Moreover, besides

time-related factors, a spatial factor represented by latitude is also expected to affect the model buildup. The first aim of this study is to analyze how ML models treat input variables related to temporal (peak season vs whole year) and spatial variability.

The second aim is to use different ML models to understand how the best model compares to process understanding of the carbon fluxes' dynamics. In addition to that, we aim to compare different ML models and check if all of them reproduce $CO_2$ flux dynamics robustly if they tend to choose the same important input variables, and if dependences on these variables are

similar between the models.





Finally, we combine data sets from two different latitudes, include data from a post-thinning period in Hyytiälä, and use XAI to understand how the models perform when different ecosystems exist in the same data set. We introduce additional variables distinguishing between the sites and model NEE with and without these variables.

In this study, we focus on several research questions: 1) compare the ML models' performance for two ecosystems from
different latitudes but with the same main tree species using accuracy metrics and XAI (with a linear model as a baseline); assess the reliability of results based on the robustness of their reproduction by different models; 2) analyze the shift in the choice of model variables and their general performance depending on the seasonality (i.e., peak growing season or the whole year) and latitude; 3) study how combining the data sets from the two studied forest ecosystems at different latitudes and including post-thinning data affects model results.

## 2   Materials and methods

### 2.1   Stations and data sets

We used atmospheric observations from the SMEAR I station in Värriö, Finland (Hari et al., 1994), and the SMEAR II station in Hyytiälä, Finland (Hari and Kulmala, 2005). The stations are located in the boreal forest in central Finland (Hyytiälä: 61°51'N, 24°17'E, 80 m a.s.l.) and in Finnish subarctic region (Värriö: 67°46'N, 29°36'E, 180 m a.s.l.). The mean annual air temperature
is $3.5^{o}C$ in Hyytiälä and $-0.5^{o}C$ in Värriö (source: ICOS database). The mean annual precipitation in Hyytiälä is 710 mm, and in Värriö, it is 601 mm. Forest stands at both sites are dominated by 60-65-year-old Scots pines (*Pinus sylvestris* L.). However, the average tree height differs, being ca. 10 m at SMEAR I and 19.9 m at SMEAR II, as measured in 2023. The forest canopy at SMEAR II is closed, and at SMEAR I, it is open. Both sites are part of the Integrated Carbon Observation System (ICOS) and Integrated European Long-Term Ecosystem, critical zone, and socio-ecological Research (eLTER) networks. ICOS provides
high-quality data on greenhouse gas concentrations and carbon fluxes between the atmosphere, Earth, and oceans. Conversely, eLTER focuses on long-term, site-based ecosystem research across various European locations. Each site contributes by measuring carbon fluxes and studying ecosystems, respectively. Various meteorological variables and radiation are also measured routinely at the stations. The data is publicly available to download from the SmartSMEAR database (https://smear.avaa.csc.fi/, accessed September 2022; latest updated data sets can be found at https://etsin.fairdata.fi/datasets/SmartSMEAR).

Data from Hyytiälä was divided into two separate data sets, pre-thinning, referred to just as Hyytiälä data (prior to 2019), and post-thinning (post 2019), referred to as post-thinning Hyytiälä data. The separation is due to the thinning of the forest at Hyytiälä station in 2019, the removal of smaller trees from the forest understory in 2019, and thinning (from below) in January-March 2020. In the thinning, 30% of tree basal area was removed (Aalto et al., 2023), which significantly changed NEE due to the decrease of biomass. The data set thus had too large differences to be treated as a direct continuation of the
pre-thinning data set. The amount of data points and the dates for each data set can be seen in Table 1.

The data used in this study was at a 30-minute measurement interval. The higher frequency enables a more detailed study of the daily cycle of NEE. It allows for the analysis of the impact of such variables that affect the ecosystem processes on a short time scale, such as the impact of changes in radiation on photosynthesis. Raw data for the target variable (NEE) being modeled





**Table 1.** Amount of observations and date ranges of each data set.

| Site and case | Dates | $N$ obs., train | $N$ obs., test |
|---|---|---:|---:|
| Hyytiälä, all season | 07/2008 - 09/2018 | 31275 | 7821 |
| Hyytiälä, peak | Jul-Aug (2008 - 2018) | 9382 | 2348 |
| Post-thinned Hyytiälä, all season | 02/2019 - 05/2021 | 8767 | 2923 |
| Post-thinned Hyytiälä, peak | Jul-Aug (2019 - 2020) | 1032 | 344 |
| Värriö, all season | 05/2013 - 10/2019 | 20908 | 5230 |
| Värriö, peak season | Jul-Aug (2015 - 2019) | 5736 | 1436 |

with the machine learning models is first captured using eddy covariance technique (Aubinet et al., 2012) and then processed
to NEE using the EddyUH software (Mammarella et al., 2016). Negative NEE corresponds to the ecosystem acting as a net
carbon sink, while positive corresponds to the ecosystem acting as a net carbon source. We model NEE using meteorological
variables such as air temperature, soil temperature, solar radiation, relative humidity, and soil moisture content. LAI is not used
here as its seasonal variability in the chosen period is relatively small (Hyytiälä - about 30%, Värriö - 20%), which translates
to below 10% change in canopy light interception and roughly the same percentage in GPP. For some input variables, minor
differences exist in how the data is measured at the two stations (e.g., soil moisture is from slightly different depths). The data
used was non-gapfilled to avoid the influence of models typically used for gapfilling. At Hyytiälä, photosynthetically active
radiation (PAR) was not measured before 2009, and we used global radiation multiplied by the PAR quantum efficiency of 2
$\mu$mol s$^{-1}$ W$^{-1}$ (Ross and Sulev, 2000; Ezhova et al., 2018) to calculate missing values of PAR. A list of all variables used can
be seen in Table 2.

In the pre-processing of the data, time points that contained missing values of any studied input variable were discarded.
Also, all rows where the PAR value was less than 10 $\mu$mol s$^{-1}$ m$^{-2}$ were filtered out due to the interest being solely on
modeling daytime NEE. We calculated the diffuse fraction:

$$F_{dif} = \frac{\text{PAR}_{dif}}{\text{PAR}}, \tag{1}$$

and vapor pressure deficit (Monteith and Unsworth, 2013):,

$$\text{VPD} = e_s - e_a, \text{ where } e_s = 611 \exp(\frac{17.27 T_{air}}{237.7 + T_{air}}), e_a = e_s \frac{\text{RH}}{100}. \tag{2}$$

In eq. (2), $T_{air}$ is in units [$^o$C] and $e_s$, $e_a$ are in units [Pa].

The machine learning models were trained using three different setups. First, the models were trained by using data from
entire years from individual sites; second, by using data of peak growth seasons (July and August) from individual sites; and
finally, by combining all the different data sets into a single mixed data set (referred hereafter as "mixed data set") and adding



**Table 2.** List of input variables used for model training.

| Abbrevation | Name | Units | Notes |
| --- | --- | --- | --- |
| PAR | Photosynthetically Active Radiation | $\mu$mol s$^{-1}$ m$^{-2}$ | Hyytiälä: Measured at 18 m height (radiation tower 12/2009-2/2017) or 35 m height (35 m tower 2/2017-). Värriö: - . |
| PAR$_{dif}$ | Diffuse PAR | $\mu$mol s$^{-1}$ m$^{-2}$ | Hyytiälä: Measured at 18 m height (radiation tower 12/2009-2/2017) or 35 m height (35 m tower 2/2017-). Värriö: - . |
| $F_{dif}$ | Diffuse Fraction | - | $F_{dif} = \frac{\text{PAR}_{dif}}{\text{PAR}}$ |
| AirTemp | Air Temperature | $^{o}$C | Hyytiälä: Measured at 33.6 m height. Värriö: 9 m |
| SoilTempA | Soil Temperature | $^{o}$C | Hyytiälä: Measured 2-5 cm depth in the mineral soil). Värriö: 5cm. |
| SoilTempB | Soil Temperature | $^{o}$C | Hyytiälä: Measured 22-29 cm depth in the mineral soil (Only in Hyytiälä) |
| VPD | Vapor Pressure Deficit | Pa | Formula listed section 2.1 |
| SoilWatCont | Soil Water Content | % | Hyytiälä: 26-36 cm depth in the mineral soil. Värriö: - . |
| RH | Relative Humidity | % | Hyytiälä: Measured at 16 m height (4/1998-1/2017) or 35 m height (2/2017-). Värriö: 2m. |
| FricVel | Friction Velocity | m/s | Hyytiälä: Measured at 24 m height, 27 after 2019. Värriö: Measured at 16.6 m height |

a label that denotes from which site the data originates from ('Värriö', 'Hyytiälä' for Hyytiälä pre-thinned, 'HyytiäläT' for Hyytiälä thinned).

## 2.2   Machine learning models

To ensure robustness and reduce potential biases, we validate our findings across four distinct ML models, aiming to identify consistent patterns or insights and provide an overall picture of how well the models can use this data to predict NEE. Applying
several models to the same data set provides a context on what input variables are consistently considered important across different models. The four models used were Cubist (Quinlan, 1992), Random Forest (Breiman, 2001), avNNet (Kuhn, 2008), and basic Linear Regression (Kutner et al., 2004). All were implemented in R (v. 4.3.0: https://www.r-project.org/) using R's "caret" library (v. 6.0.94: https://github.com/topepo/caret/). Linear Regression served as the baseline model, while the other models were chosen due to their proven competence in solving various regression problems (Fernández-Delgado et al., 2019).





Random Forest is a popular model that has been used in previous research (Cai et al., 2020; Liu et al., 2021; Abbasian et al., 2022; Zhu et al., 2023) due to its ease of use, high accuracy, and robustness. It is an ensemble model that uses the averaged output of random regression trees (Fernández-Delgado et al., 2019) by training different regression trees on different subsets of the data. The final prediction is the average result of the different tree predictions. The algorithm is quite robust as the different trees are trained with the different subsets of the training data. The randomForest library (Liaw and Wiener, 2002) implements
the regression algorithm of Random Forest used in this study.

Cubist is one of the best-performing regression models (Fernández-Delgado et al., 2019) across multiple types of data sets (i.e., type and size of data). Like Random Forest, it is created from multiple individual regression trees, where each terminal leaf contains a smoothed linear regression model for prediction (Zhou et al., 2019). It creates a series of "if-then" rules that can be considered the branches of a tree, while the leaves are an associated multivariate linear model. The corresponding model is used
to calculate the final predicted value as long as the set of covariates satisfies the conditions of the corresponding rule. Cubist also uses boosting with its training committees, which creates a series of trees with different weights and nearest-neighbors search to adjust the predictions better.

Model Averaged Neural Networks (avNNet) is a single hidden layer feed-forward neural network characterized by their architecture and training approach. The network consists of interconnected neurons arranged in layers, with the final layer
outputting the prediction (Ripley, 2007). During the training phase, initial weights, which influence predictions, are randomly assigned. These weights are then iteratively updated, enabling the network to capture nonlinear relationships. Given the randomness in predictions due to these initial weight assignments, avNNet constructs multiple neural network models and averages their results. This averaging process promotes a more robust and stable prediction by minimizing the impact of any single model's randomness.

The basic multivariate Linear Regression is used as a baseline to understand how much impact and improved results more advanced models can provide. Linear regression finds a linear relationship between the independent and dependent variables determined by minimizing the sum of the squared differences between the predicted and the actual values(Hastie et al., 2009).

## 2.3   Cross-validation framework, Hyperparameter tuning and validation metrics

$K$-fold cross-validation is a resampling method for validating model efficiency, which generally results in less biased mod-
els (Jung, 2018). $K$-fold cross-validation method shuffles the data set randomly and splits it into $K$ groups or folds. First, each fold is taken as a holdout, while the model is fit on the rest of the folds, and then the model is evaluated on the holdout set. The score is retained, and the model is discarded. In repeated $K$-fold cross-validation, this process is done $R$ times on different splits.

During the model training, repeated $K$-fold cross-validation was used with Caret librarys (Kuhn, 2023) grid hyperparameter
search. This method trains and evaluates a model using all possible combinations of specified hyperparameter values to identify the combination that yields the best model performance. It was used to tune the models' hyperparameters and configuration settings that are external to the model and can be adjusted to optimize performance. Values $R = 5$ repeats and $K = 10$ folds were used to fit each model. The tuned hyperparameters can be seen in Table 3. The train and test data as well as the folds of the



**Table 3.** List of the final model hyperparameters with their respective values for each data set. Values correspond to different sites: Hyytiälä All, Hyytiälä Peak, Värriö All, Värriö Peak, All Site Models with site label and All Site Models without Site label.

| Method | Hyperparameter | Description | Values |
|---|---|---|---|
| Cubist | committees | Number of committees (models) to be fitted. | 100, 90, 100, 100, 100, 100 |
| | neighbors | Number of nearest neighbors used in prediction. | 9, 9, 6, 3, 9, 6 |
| Random Forest | mtry | Number of variables sampled at each split. | 3, 3, 6, 2, 13, 9 |
| | min node size | Minimum size of terminal nodes (leaves). | 5, 5, 5, 5, 5, 5 |
| avNNet | size | Number of units in the hidden layer(s). | 13, 13, 13, 13, 13, 13 |
| | decay | Weight decay parameter for regularization. | 0.1, 0.1, 0.1, 0.1, 0.1, 0.1 |
| | bag | Boolean flag for using bootstrap aggregating (bagging). | False, False, False, False, False, False |

$K$-fold cross-validation were split using a predetermined random split to ensure repeatability. However, due to technical limitations, in-depth hyperparameter tuning was not used on the models that contained data from all sites. Instead, hyperparameters based on the results from the single-site models were used.

In evaluating the performance of our machine learning models, we primarily relied on two key metrics to assess the models' goodness of fit: the coefficient of determination ($R^2$) and the root mean squared error (RMSE). RMSE measures the differences between the values predicted by a model and the actual values and provides an understanding of the magnitude of error the model might make in its predictions. A lower RMSE indicates a better fit to the data, implying that the model's predictions are more precise. The models' hyperparameters were tuned specifically based on the RMSE score.

## 2.4 Explainable AI Methods

As machine learning models have been used more in research and industry, the demand for more transparent and interpretable models has grown (Dwivedi et al., 2023). As the model accuracy has risen, so has the model complexity. The highly accurate and complex models have many hyperparameters that can not be made human-understandable. To be trustable, the ML model must produce interpretable or transparent results. Relying on unexplained or inaccurate predictions can lead to critical errors. Accuracy metrics do not always portray the true prediction capability of a model, so it is vital to critically evaluate the results against existing knowledge or theories. XAI methods aim to provide machine learning models and methods that enable users to better understand, analyze, and evaluate the models' decision-making.

In this study, we used two XAI methods: Accumulated local effect (ALE) plots and permutation feature importance (Molnar, 2020). They provide insight into how the input variables affect a model's output. Both are model-agnostic global methods, meaning they can be used regardless of the selected model and provide interpretations on the data set as a whole rather than individual points (Molnar, 2020). Both of these methods were implemented using R's "iml" library (v.0.11.1: https://github.com/christophM/iml/, Mol- nar et al. (2018)).





### 2.4.1 Permutation Feature Importance

Permutation feature importance is a method that aims to measure the increase in the prediction error of a model after the input variables (features) are permuted. In permutation feature importance, the relationship between a specific input variable and the variable the model tries to predict is deliberately disrupted to understand how the models' prediction accuracy is affected (Molnar, 2020). If an input variable is important, randomly rearranging its values increases the model error, as the model then relies on that specific input variable for an accurate prediction. Trained model is denoted as $f$, input variable matrix as $\mathbf{X}$, target vector as $\mathbf{y}$, and error measure $L(\mathbf{y}, f(\mathbf{X}))$. The algorithm works as follows:

1. Estimatse the original model error $e = L(\mathbf{y}, f(\mathbf{X}))$

2. For each input variable with index $i \in \{1, ..., p\}$, where $p$ is the total number of input variables, the following is done:

   2.1 Generates a permutated input variable matrix $\hat{\mathbf{X}}$ by permuting input variable $i$ in the data $\mathbf{X}$, which breaks the association between input variable $i$ and the true outcome $\mathbf{y}$.

   2.2 Estimates the error caused by the permutation by predicting with it $\hat{e} = L(\mathbf{y}, f(\hat{\mathbf{X}}))$.

   2.3 Calculates permutation input variable importance as quotient $Imp_i = \hat{e}/e$.

3. Sorts input variables by descending $Imp$.

Only test data is used to calculate the permutation feature importance. Assessing feature importance using the training data might result in too inflated scores due to a model overfitting on training data. That said, the features with very high scores might not be as important for making accurate predictions on new, unseen data.

### 2.4.2 ALE Plots

Accumulated local effect (ALE) plots describe how input variables influence the prediction of a machine learning model on average (Molnar, 2020). ALE reduces a complex machine learning function to a function that depends on only one, as in our case, or two input variables and visualizes the effects between a selected variable and the prediction of the target variable of a machine learning model. The idea is to remove the unwanted effects of other input variables, take partial derivatives (local effects) of the prediction function with respect to the feature of interest, and integrate (accumulate) them with respect to the same feature.

The value of ALE at a certain point can be thought of as the effect of the selected variable at a specific value compared to the average prediction made on the data. To calculate ALE value for input variable $s$ at point $x \in [min(\mathbf{x_s}), max(\mathbf{x_s})]$, with $\mathbf{x}_s$ being the vector of this variables values, the input variable values $\mathbf{x}_s$ are divided into $K$ intervals, where the start of the first interval is the lowest value $z_0 = min(\mathbf{x}_s)$, and the differences of predictions between the sequential intervals is calculated. While the exact ALE formula requires a model with a derivative, an approximate version is used here that is more widely adopted and works for models without a derivative. Initially, an uncentered effect is computed:





$$\bar{f}_{s,ALE}(x) = \sum_{k=1}^{k_s(x)} \frac{1}{n_s(k)} \sum_{i:\ x_s^{(i)} \in ]z_{k-1,s}, z_{k,s}]} \left[ f(z_{k,s}, \mathbf{x}_{-s}^{(i)}) - f(z_{k-1,s}, \mathbf{x}_{-s}^{(i)}) \right].$$

The input variable of interest is replaced with grid values $\mathbf{z}$, where the grid values represent the edges of the intervals. The interval index an input variable value $x \in \mathbf{x}_s$ falls in is denoted as $k_s(x)$, while $n_s(k)$ denotes the number of observations inside the $k$-th interval of $\mathbf{x}_s$. A single data point is denoted as $\mathbf{x}^{(i)} = (x_s^{(i)}, \mathbf{x}_{-s}^{(i)})$, where $x_s^{(i)}$ denotes the $i$-th value for the selected input variable, and $\mathbf{x}_{-s}^{(i)}$ is the vector of all the other features of a single data point that are kept constant. The ML predicting function is denoted as $f$.

The differences between the predictions $f(z_{k,s}, \mathbf{x}_{-s}^{(i)}) - f(z_{k-1,s}, \mathbf{x}_{-s}^{(i)})$ are the effect that the input variable $s$ has for an individual data point to predicting the dependent variable (NEE in our case) when using the upper and lower values of an certain interval. The sum $\sum_{i:\ x_s^{(i)} \in ]z_{k-1,s}, z_{k,s}]}$ adds up the effects of all instance within an interval $x_s^{(i)} \in ]z_{k-1,s}, z_{k,s}]$. This is then divided by the number of observations in this interval $n_s(k)$ to obtain the average difference of the predictions of this interval. The sum $\sum_{k=1}^{k_s(x)}$ accumulates the average effects across all intervals, meaning that the uncentered ALE of an input variable of interest is accumulated by all its previous intervals. After that, the effect is centered, making the mean effect zero:

$$f_{s,ALE}(x) = \bar{f}_{s,ALE}(x) - \frac{1}{n} \sum_{i=1}^{n} \bar{f}_{s,ALE}(x_s^{(i)}).$$

The value of ALE can be thought of as the main effect of the input variable at a certain value compared to the average prediction of the data. ALE plot has the advantage that it generates valid interpretations even if the variables are correlated, an issue that persists in other methods that reduce a prediction function $f$ to a function that depends on a single input variable such as PDP or M-plots (Molnar, 2020). As with permutation feature importance, only the test data set was used to reduce the chance of inflating scores due to a model overfitting on the training data set.

## 3    Results and discussion

### 3.1    NEE modelling for Hyytiälä and Värriö data sets

In this section, we report the results obtained with different models for the two cases: Värriö data and pre-thinning conditions in Hyytiälä. The results are shown separately for the whole year and for the peak growing season. First, we assess models' performance with routinely used accuracy metrics (R-squared, coefficients of determination, and root-mean-squared-error, RMSE), visualize diurnal/annual NEE cycles, and then use XAI methods. In each subsection, we start the discussion with the peak growing season results and continue with the all-season results.

### 3.1.1    Assessing model performance using accuracy metrics

Figs. 1 and 2 show coefficients of determination and RMSE, respectively, for all the models, two stations, and trained on the data sets for the whole year and the peak growing season. In general, the models perform better if trained on the Hyytiälä data set



compared to the Värriö data set, as seen from higher R-scores. If the model is used on the training data set, the R-coefficients and RMSE are somewhat better than when used on the test data set, as expected. This effect is especially pronounced for Random Forest, reaching high scores (>0.9) in all cases on the training data, but also for Cubist. The difference between the train and test scores is larger for Värriö than for Hyytiälä. Neural network and Linear models have almost identical scores on training and test data sets. In what follows, the results are reported for the test data sets if not stated otherwise.

For the peak growing season, all four models perform well, including linear regression, which is only slightly worse than the more complex models. For Hyytiälä, all nonlinear ML models give similarly high $R^2$, close to 0.9, and RMSE values almost do not differ between these models. For Värriö, RF is slightly better than other ML models, demonstrating both higher R-squared and lower RMSE. Compared to Hyytiälä's $R^2 = 0.88$, Värriö's R-squared is smaller, $R^2 \simeq 0.75$, which could be related to the presence of a larger relative number of outliers in the data or a smaller variability range of a predicted variable. The predictors vary within similar ranges in Hyytiälä and Värriö, whereas the predicted variable NEE has a larger value range in Hyytiälä compared to Värriö (corresponding to a weaker carbon sink in Värriö) because Värriö ecosystem is less productive. Similar decrease in R-scores for the cases when predicted variable had a smaller variability range was reported by Liu et al. (2021) and Abbasian et al. (2022). The difference in R-scores could be because the chosen predictors have a more significant effect on forest carbon balance in Hyytiälä than in Värriö. A big proportion of Värriö data comes from the dormant season, when the explanatory variables have virtually no effect, and the fluxes are near zero all the time. In addition, the temperature effect on NEE in Värriö could be more complicated than in Hyytiälä, with a stronger accent on a temperature history (Mäkelä et al., 2004): ecosystem response to temperature has a delay which is only partially accounted here by the use of soil temperature as one of the input variables. It is also known that Värriö ecosystem responds weakly to cold spells during the growing season, making temperature dependence even more complex. Note that also process-based models experience difficulties in reproducing carbon fluxes at sites with low leaf area index (Mäkelä et al., 2019).

Scatter plots of measured vs. modeled data for training and test data sets are shown in Fig. 3 using one of the best performing models, RF. The lowest values of NEE tend to be overestimated, and the largest ones underestimated. This is seen best in the training data sets (because they are much larger) deviating from 1:1 lines at the extremes of the data. In that sense, higher scores of the training data sets should not be deceiving: a high correlation does not mean that the model values correspond perfectly to the measured values. In Fig. 2, it is visible that RMSE values for Värriö are smaller than those for Hyytiälä, which means that Värriö values in Fig. 3 are closer to the best-fit lines. Still, again, it does not mean that the model is better because the best-fit line of the measured vs. modeled data points is not 1:1. By high accuracy scores, the mean diurnal cycle of NEE within the peak growing season is almost perfectly reproduced by the RF model (Fig. 4) with slightly smaller standard deviations in the modeled than measured data.

In the case of the all-season data sets, the performance of linear regression drastically decreases when compared to the peak growing season data sets (Figs. 1, 2). This could be expected because, on the whole-year scale, NEE dependence on many variables becomes nonlinear. Especially for the Värriö data set, linear regression scores fall below 0.5, and RMSE increases by 40% compared to nonlinear ML models, meaning that more complex models are needed and justified. Figs. 3 - 5 show scatter plots and annual daytime NEE cycle for Hyytiälä and Värriö. The same conclusions as for the peak growing season data sets





apply here as the mean values were almost perfectly reproduced and extreme values missing. The models captured the essential features of the annual NEE cycle, including ecosystem spring and autumn phenological transitions (Fig. 5).

It is interesting to consider different models' performance for the same setup. Here we show an example for Hyytiälä all-year data set (Fig. 6). The test cases for all ML models look similar. Note orange points (test RF) covering black points (training

RF) illustrating the smaller RMSE for the training data set. The linear model plot is more scattered, and the points are not organized along one line (in agreement with previously reported low R-scores and high RMSE).

Concerning other studies, Dou and Yang (2018) demonstrated that in modeling whole-year NEE of forest ecosystems, the $R^2$-scores as high as 0.64-0.84 can be reached on the test data sets for separate ecosystems. Our scores are within this interval for Värriö and significantly higher (0.90) for Hyytiälä. However, we used a different, more diverse set of input variables and

modeled half-an-hour fluxes compared to daily fluxes in the study mentioned above.

### 3.1.2 Feature importance and ALE

Now, we consider feature importance and ALE, which allows us to analyze how the model treats the data sets. For the peak growing season, all nonlinear ML models agree for both stations (Fig. 7, Table A1) that the variables with the most explanatory power are PAR and the diffuse radiation. Moreover, PAR typically comes first, except for Cubist in the case of Värriö. The third

variable in importance is vapor pressure deficit (3 cases), air temperature (2 cases), or soil temperature A (1 case). It is good to note that VPD is calculated based on air temperature (see Sec. 2.1), so these variables are not independent. These variables also have a similar effect increasing NEE at high temperatures: due to increasing respiration (driven by temperature increase) and due to dampening photosynthesis (driven by VPD increase). Interestingly, the linear model chooses diffuse radiation as the most important variable to explain NEE, likely because the dependence of photosynthesis on diffuse radiation is closest to

linear. PAR is theoretically the most important variable during the peak growing season to explain photosynthesis (Palmroth and Hari, 2001). Light response curve, quantifying the dependence of $CO_2$ flux due to photosynthesis on PAR, saturates at higher PAR values. Respiration does not depend on PAR, but it grows exponentially with temperature. Therefore, during the peak growing season, we can expect the relationship of NEE on PAR to be similar to that of photosynthesis, saturating at higher PAR values.

Overall, during the peak growing season in boreal forests, daytime $CO_2$ flux due to photosynthesis prevails over that due to respiration, at least in Hyytiälä (Kolari et al., 2009). Therefore, one can expect that light control on photosynthesis also dominates the NEE response, and the models consider light-related variables as the most important. At the same time, the temperature effect is relevant for respiration, and VPD controls stomatal conductance and carbon uptake. Accordingly, the models pick temperature or VPD as the third important variable.

ALE demonstrates that NEE decreases with increasing PAR and diffuse radiation for all models (Fig. 8). Nonlinear models capture the nonlinear dependence of NEE on PAR, which is most pronounced for the RF model. As captured by nonlinear models, NEE response to diffuse radiation is also relatively strong. However, for the Värriö data set, NEE levels off at the largest diffuse radiation. High diffuse radiation level, observed under a cloudy sky, means in Värriö that the PAR level can be below the light saturation point (Ezhova et al., 2018), and therefore NEE increases. RF and Cubist also capture a nonlinear



dependence of NEE on the vapor pressure deficit, which has an optimum value between the low and high values of VPD. At very high VPD, stomatal closure prevents plants from losing water (Running, 1976), affecting again also photosynthesis. At low VPD, when water vapor pressure at the leaf level and in the atmosphere is about the same, there is no driving force to sustain transpiration. This inhibits water uptake by the roots and generally slows down plant metabolism, also affecting photosynthesis. PAR, diffuse PAR, and VPD are confirmed as essential drivers of carbon assimilation in numerous studies on

photosynthesis in different ecosystems (Gu et al., 2002; Larcher, 2003; Grossiord et al., 2020). Particularly for Hyytiälä during the growing season, a statistical model showed that daily photosynthesis is most sensitive to light and VPD (Peltoniemi et al., 2015). Note that dependencies of NEE on PAR, diffuse radiation, and VPD are qualitatively similar in all studied models, though quantitatively, sensitivity to the corresponding variables somewhat differs. However, the dependence of NEE on air temperature is not the same in all models. RF and Cubist feature increase of NEE with air temperature, whereas linear model

and artificial neural network demonstrate a decreasing dependence. In Värriö, all models except the artificial neural network suggest an increasing dependence of NEE on air temperature. Increasing dependence is in line with the stomatal control at high temperatures (stomatal closure dampening photosynthesis) and higher soil respiration.

It is interesting to analyze ALE from different models trained on the input data sets with several temperature variables. Both soil and air temperature are typically included in modeling studies based on machine learning (Dou and Yang, 2018;

Liu et al., 2021; Abbasian et al., 2022). Cai et al. (2020) and Wood (2021) include average, minimum, and maximum air and soil temperature in their studies, adding more interdependent variables in the data sets. Hyytiälä's data set includes air temperature and temperature from A and B soil horizons. In the peak season, all these temperature-related variables show quite similar dynamics. With soil depth, the mean temperature and amplitude of the diurnal temperature cycle decrease, and the time lag between the temperature signals increases. The time delay between air temperature and soil temperature at B

horizon is generally smaller than half a day. All the models, besides RF, treat Soil temperatures A and B as important variables and demonstrate strong but opposite dependencies on these variables (Fig.8). Keeping in mind that soil temperatures A and B are correlated (Appendix A, Fig. A1), opposite NEE dependencies may outweigh each other. Strong opposite dependencies on correlated variables should be treated cautiously. The models might use them to compensate for the effect on NEE from different temperature variables or for tuning towards higher scores on given data sets. In that case, there is no guarantee that

this compensation or tuning will work for a higher temperature, which is currently not represented in the data set. The same conclusion applies to using the model developed for a particular site on the data sets from other sites (Peltoniemi et al., 2015). In contrast, RF shows a strong increasing dependence only on air temperature and a weak dependence on two soil temperature variables.

Regarding other variables that have a more minor effect on NEE:

- diffuse fraction has a similar effect for all the models: increasing NEE with increasing diffuse fraction. This is likely due to a large diffuse fraction under an overcast sky when radiation is low, and photosynthesis inhibited.

- soil temperature A: All models besides RF show NEE increase with temperature in Hyytiälä and decrease in Värriö. Only in the case of RF does NEE consistently increase, but the absolute effect of this variable on NEE is relatively small. An increase in NEE is expected for higher temperatures, primarily due to the respiration effect.




- relative humidity: VPD is also linearly dependent on this variable. The higher the RH, the closer ambient air is to saturation, and VPD, in this case, is small. Low RH, vice versa, favors higher VPD values. This is captured better by RF and Cubist than by the neural network and linear model. Having VPD as one of the powerful explaining variables should, in principle, diminish the role of RH. However, this is not the case for neural network and linear models. RH is placed relatively high in the feature importance for these models, which is reflected in the significant range of NEE variability due to RH.

- soil water content: In Hyytiälä, all models feature an increase in NEE for low water content. In Värriö, all models feature an increase of NEE with growing water content, and similar behavior is demonstrated by Cubist and neural network in Hyytiälä. Note, however, that sensitivity to this variable is quite low for all models, indicating that soil moisture does not limit ecosystem functioning in current conditions. However, this could change in the future, which would perhaps not be captured by the models.

- friction velocity: In Hyytiälä, all models show a similar decreasing dependence of NEE on this variable, meaning that NEE flux is somewhat sensitive to the level of turbulence. On the one hand, this could indicate an eddy covariance problem (Moffat et al., 2010). On the other hand, this dependence might reflect physical processes: friction velocity has a weak increasing trend in Hyytiälä due to trees getting taller, which coincides with the weak, increasing trend in carbon uptake but not in respiration (Launiainen et al., 2022). In Värriö, there is no clear dependence. In general, this variable is also not important.

Feature importance for the all-season data (Fig. 7) shows another set of most relevant variables bringing up soil temperature at the expense of air temperature or vapor pressure deficit (nonlinear models, see Table A1). In many cases, soil temperature becomes the second important variable, sometimes even the first (neural network, Hyytiälä). The increasing importance of the temperature-related variable is expected because, in the all-season case, the model needs to capture the seasonality of carbon flux (Mäkelä et al., 2004, 2006), and soil temperature grows in summer and decreases in winter. However, the models' choice

of soil temperature over air temperature requires additional explanation. Presumably, the soil temperature is positive during the warm season and nearly constant during the snow season. This behavior is in line with NEE, which is also inhibited in winter. Air temperature, in contrast, may display significant variability. In addition, soil temperature limits plant water use and photosynthesis in spring and autumn (Wu et al., 2012; Lintunen et al., 2020). In the case of the linear model, PAR is no longer among the three most important variables, replaced by another soil temperature or the diffuse fraction of radiation.

The dependence of NEE on light variables (PAR, diffuse radiation) remains largely similar to that for the peak growing season setup (Fig. A3). Nonlinear models predict that the NEE dependence on air temperature has a minimum in Hyytiälä in the presence of negative temperatures in the data set, suggesting larger NEE during the cold season and periods with the warmest soil. NEE has similar dynamics for soil temperature A. This might reflect the absence of GPP in the cold season and the increase of respiration for high temperatures. In Värriö, NEE dependencies on air temperature also generally have minima.

Dependence on soil temperature A is decreasing. Note that for Hyytiälä, NEE dependencies on soil temperatures B and A are again of opposite sign for all models besides RF. The linear model fails on the Hyytiälä data set featuring lower NEE and strong carbon sink at low, even negative temperatures. The failure of the linear model on the whole season cycle data could be related to its inability to capture essentially nonlinear air temperature dependence, which becomes important on the all-season scale.



Considering less critical variables, the dependencies remain mainly the same. In some cases, however, the neural network
demonstrates dependencies inconsistent with expected behavior, i.e., featuring a strong carbon sink at high air temperature
in Hyytiälä or under low RH conditions. It is worth mentioning that the dependence on soil water content has become quite
complicated in Hyytiälä, with a minimum and a maximum. This could be related to data containing subsets with high water
content at low temperatures when photosynthesis is inhibited, e.g., during snowmelt or late autumn. In any case, as for the peak
growing season setup, the sensitivity of NEE to this variable is low.

Finally, if the most important input variables for these sites are the same, and dependencies of NEE on these variables are
similar as in the case of RF and, to a lesser extent, Cubist, one could expect that it is possible to build a more generic model,
which would be able to give reasonable results for many different boreal forest sites. We, therefore, built one model based on
all the data in the following section.

### 3.2  NEE modelling: mixed data set

In this section, we report the results of NEE modeling using all three available data sets, which consist of pre-thinning Hyytiälä
(referred to just as Hyytiälä), Värriö and post-thinning Hyytiälä. We aimed to understand how the models perform in the
following cases: 1) mixed data set, containing data from both sites, including post-thinning Hyytiälä, without any separation;
2) mixed data set, but we introduce variables, which could enable the model to identify the site. This variable was initially a
categorical variable denoting the data set (either Hyytiälä, Värriö, or post-thinning Hyytiälä). However, it was transformed into
three separate binary (either 0 or 1) variable columns denoting which site the data instance is from because it is more efficient
in computation.

#### 3.2.1  Assessing model performance on a mixed data set using accuracy metrics

The determination coefficients for mixed data sets are shown in Fig.9, separately for the model runs with and without the
variables for the site identity. Adding site variables to the data set slightly improves the correlation coefficient $R^2$, which
remains high for the best models, RF and Cubist (0.84-0.87 for the peak season, 0.86-0.89 for the whole season). The results
are marginally worse for the peak growing season. Comparing this result to the results for the separate stations (Fig. 1), we
note that the scores are closer to those for Hyytiälä. This is likely because Hyytiälä data prevail in the compiled data set.
Interestingly, the linear model performs worse than others on a compiled data set, even during peak growing season. The
scores for linear models are also significantly lower than Hyytiälä scores (drop from 0.76 to 0.68).

As said, site variables do not have a significant effect on R-squared but the advantage is more evident for RMSE (Fig. 10). In
general, RMSE for the peak growing season data is larger than for the whole season. This is likely because the fluctuations and
errors of NEE measurements outside the growing season are relatively small. In addition, the flux random error increases with
the flux magnitude within the growing season. If we compare the models for the peak growing season with and without site
variables, we see that adding site variables reduces RMSE from about 2.4 to 2.15 $\mu$mol s$^{-1}$ m$^{-2}$ in the case of best-performing
models. For all season data sets, the reduction of RMSE is from about 1.8 to 1.6 $\mu$mol s$^{-1}$ m$^{-2}$. Considering models for the
separate sites, there is a slight reduction in RMSE score in the model, including site variables and all-season data set, compared





to Hyytiälä. Overall, the site variables barely improve the correlation between measured and modeled points but reduce the scatter in the plot presenting measured vs modeled points.

### 3.2.2 ALE and feature importance for the mixed data set

It is interesting to assess the position of the site variable in the feature importance diagrams (Fig. 11). Consider first the peak growing season. Note that the vapor pressure deficit is no longer within the three most important variables for the RF model and the mixed data set. It is replaced by the soil water content for the setup without site variables and by the 'Värriö' site variable for the model with site variables. Värriö is located at higher latitudes and has different soil characteristics: soil moisture is lower there (Fig. A2). This input variable could then be considered a replacement of the site variable by the RF model. However, other

nonlinear models consider VPD the third most important variable when no site variables are introduced. Cubist treats 'Värriö' and VPD as the third-fourth important variables when the site variables are included. The neural network still considers VPD a more powerful NEE explanation variable than any site variables.

Judging by ALE (Fig. 12), the models prescribe higher values of NEE to the drier cases, which is in line with how the ecosystem functions under drier conditions (reduction of photosynthesis). When site variables are added, RF and Cubist choose

site variable 'Värriö' as the third important. From ALE, one could see that the modeled NEE increases if 'Värriö' changes from zero to one. The models then use this site variable to make all NEE values at Värriö somewhat higher than the general mean value for all three sites, which is the case due to lower tree biomass. Interestingly, the linear model does not use the site variable 'Värriö' at all (its value is zero). Similarly, models use the variable 'Hyytiälä' when it is equal to one to decrease NEE, and this decrease is less pronounced for the best-performing models. Finally, when 'HyytiäläT' variable is equal to one, RF and Cubist

slightly increase NEE, whereas the other models decrease NEE. Because the prevailing data set is still Hyytiälä pre-thinned, this data set likely dictates the base values chosen by the models. Therefore, a moderate increase of NEE for the Hyytiälä thinned data set and a stronger NEE increase for the Värriö station is reasonable.

Furthermore, for the setup with site variables to predict NEE in peak growing season, all the models display increasing dependence on air temperature, following theoretical expectations due to increasing respiration and reduced photosynthesis.

However, NEE dependencies on soil temperature A somewhat decrease for all the models except RF. Note that neural network and linear models still have much stronger NEE dependencies on VPD than RF and Cubist. Similar to setups for separate stations, the models likely use relative humidity to compensate for a too-strong effect of VPD.

Modeling the all-season data sets without site variables, we note that NEE still has the same decreasing dependence on soil water content (Fig. A4), and the dependence is quite strong, in contrast to what was observed when we modeled separate sites.

However, the feature importance order for the RF and Cubist models repeats that for single site Värriö: PAR, soil temperature A, and diffuse radiation (Fig. 11). (Recall that the Hyytiälä feature importance set contained Soil temperature B, not A. Replacement of this variable by the temperature at A horizon is because Soil temperature B is not in the data set anymore as it was not measured in Värriö). The neural network, similar to the model results for separate sites, has a strong accent on VPD and air temperature independently of the presence of the site variables in the input data set. Note that site variables are not

among the three most important in the all-season case. Instead, the models retain their dependence on soil temperature, which



allows them to reproduce a seasonal cycle. This can be interpreted as the sign that on the all-season scale, the model deems the changes in NEE due to the seasonal cycle more critical than those associated with the site-to-site change. Nevertheless, site variables appear among the six most important features in the feature importance diagrams, and as follows from Fig.10, they help to reduce the RMSE.

Further looking at the all-season setup, dependencies of NEE on PAR, diffuse radiation, and soil temperature A for all models (Fig. A4) are similar to those for separate stations (Fig. A3). RF demonstrates a somewhat stronger increasing NEE at higher temperatures than other models. All the models use the 'Hyytiälä' site variable to slightly decrease NEE and the 'Värriö' variable - to increase (besides the linear model, which still does not use the 'Värriö' variable at all).

From both ALE plots for the peak season and all-season setups (Figs. 12, A4), it follows clearly how soil water content loses
its strong position when site variables are introduced and how the NEE dependence on this variable again becomes complex, in line with what is observed for separate stations.

Generally, RF performed more in line with theoretical expectations than other models on the input data set containing interdependent variables, such as VPD depending on air temperature and humidity or several temperature variables. The linear model and the Neural network demonstrate strong dependencies of NEE on VPD, which they likely compensate for by relatively
strong dependencies on air temperature and relative humidity. In some cases, these dependencies contradict the expectations based on ecosystem functioning (e.g., strengthening of carbon sink with increasing air temperature for Hyytiälä during peak season).

## 4 Conclusions

We modeled NEE at two sites in boreal forest separately: one in central Finland and one in the Finnish subarctic. We focused
on the peak growing season and all-season data set. Peak growing season NEE for separate sites can be modeled reasonably well even with the linear model. However, the linear model performs significantly worse than nonlinear ML models in the case of several sites and all-season data sets.

The most powerful explaining variables in the peak growing season setup are PAR, diffuse PAR, and vapor pressure deficit (or air temperature); in the case of the all-season setup, such variables are PAR, diffuse PAR, and soil temperature. This is a robust
result reproduced by most of the models used in this study. High vapor pressure deficit dampens photosynthesis and, hence, makes NEE increase. This effect is essential during the peak growing season. The models presumably used soil temperature to account for the change in NEE within a seasonal cycle. Based on ALE, the models that give qualitative dependencies on different variables in line with theoretical expectations are mainly Random Forest and Cubist. This result aligns with many studies that used RF based on its best performance compared with other models. At the same time, in setups with several
interdependent variables, the Linear model and artificial Neural network often display strong opposite dependencies on the interdependent variables, largely canceling their total effect on NEE.

To build a joint model for several sites, we added site variables. The model is more sensitive to these variables within the peak growing season, whereas soil temperature retains its importance for the all-season data sets. In the mixed data setup, the





model scores seem to be governed by the most represented set, which in our case was Hyytiälä pre-thinned. In the absence of site variables, the models replace it with the variable that differs most between the sites: in our case, it is soil water content. NEE dependence on soil water content and the importance of this variable for NEE predictions change drastically for the models built on the data sets, including and excluding site variables.

Our ALE results suggest that RF and Cubist show more robust behavior modeling complex nonlinear dependence of net ecosystem exchange on the set of interconnected variables. They could qualitatively reproduce the theoretically expected dependencies of NEE on the major climatic drivers of ecosystem processes under different conditions and for several sites. At the same time, the Linear model and Neural network tend to overweigh some variables and compensate for those with the help of other interdependent variables. In our modeling study, the variables used by the Linear model and Neural network for compensation included air temperature and relative humidity, which are very sensitive to changing climate.

All in all, it should be noted that the models' performance changes depending on a given setup, so no single recommendation suggesting or prohibiting a specific model can be given. This is, instead, a case-by-case issue. Therefore, we call for broader usage of Explainable Artificial Intelligence methods when applying ML methods, especially when choosing the most suitable model. Feature importance and ALE plots together allow for a direct comparison between ML model functioning and process-based models.

Finally, we showed that even a simple way to account for the difference between the sites decreases RMSE and improves the model. The next step is to introduce a more suitable variable, allowing us to distinguish the ecosystems from each other. As Hyytiälä data are split into pre-thinned and post-thinned, we need a variable that could account for this change in the vegetation. The best candidates for this could be satellite-based NDVI and LAI (Launiainen et al., 2022; Zhu et al., 2023), which we plan to add to our data set instead of site variables.




*Author contributions.* EE, AL, KH and MK designed and conceptualized the study. TL performed modeling and prepared figures, wrote the
manuscript (Introduction and Section 2). EE interpreted results and wrote the manuscript (Introduction, Section 3 and Conclusion). AL, PK,
IM, KH and MK contributed with results interpretation, review and editing. All the authors commented on the manuscript.

*Competing interests.* The authors declare that they have no conflict of interest.

*Acknowledgements.* We acknowledge the following projects: ACCC Flagship funded by the Academy of Finland grant number 337549
(UH) and 337552 (FMI), Academy professorship funded by the Academy of Finland (grant no. 302958), Academy of Finland projects no.
1325656, 311932, 334792, 316114, 325647, 325681, 347782, "Quantifying carbon sink, CarbonSink+ and their interaction with air quality"
INAR project funded by Jane and Aatos Erkko Foundation, and HORIZON EUROPE (Project 101056921 — GreenFeedBack). University of
Helsinki support via ACTRIS-HY is acknowledged. University of Helsinki Doctoral Programme in Atmospheric Sciences is acknowledged.
Support of the technical and scientific staff in Hyytiälä is gratefully acknowledged.



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

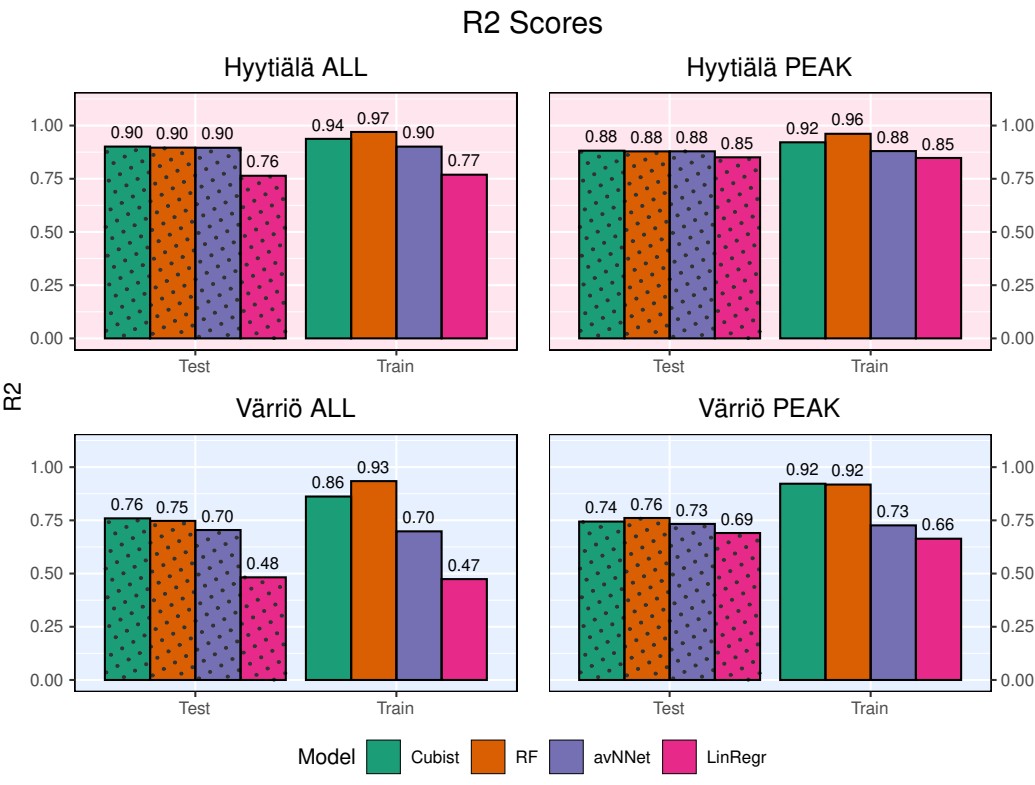

**Figure 1.** $R^2$-coefficients for all models and different data sets. In each of the four panels, the results for the training data set are shown on the right (marked 'Train'), and the results for the test data set are shown on the left (dotted bars, marked 'Test'). 'All Season' denotes the scores for the models using the whole year data sets; 'peak Season' - for the models using the peak growing season data sets.



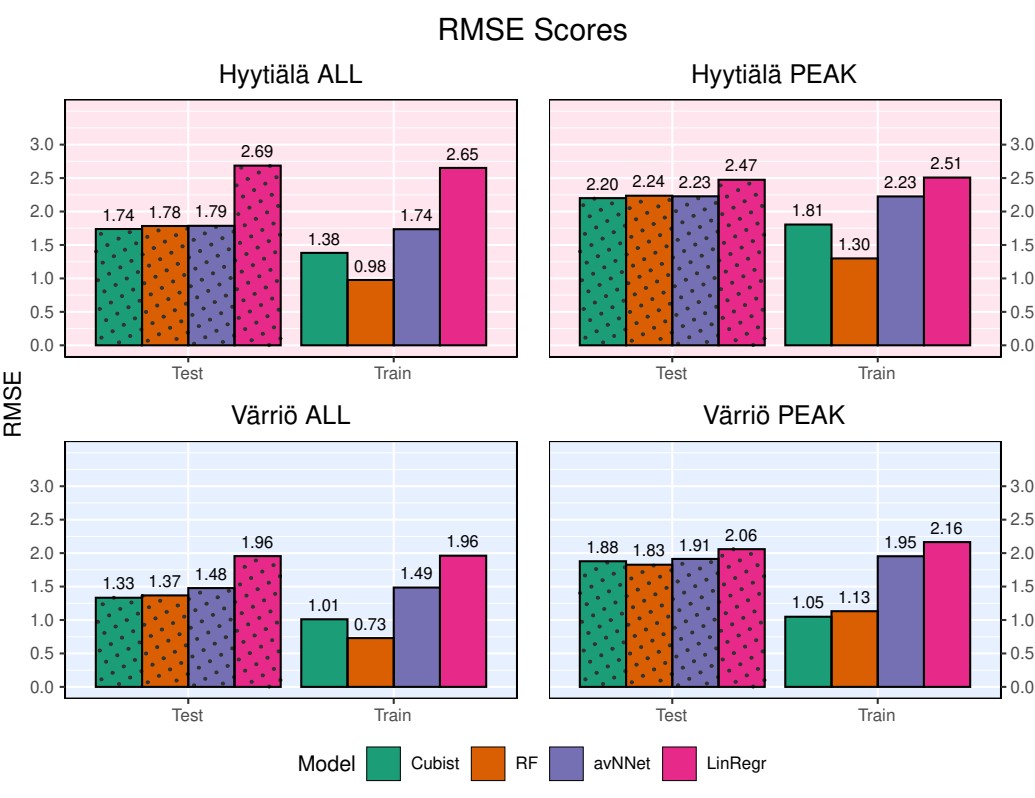

**Figure 2.** RMSE for all models and different data sets. See caption to Fig. 1 for further description.



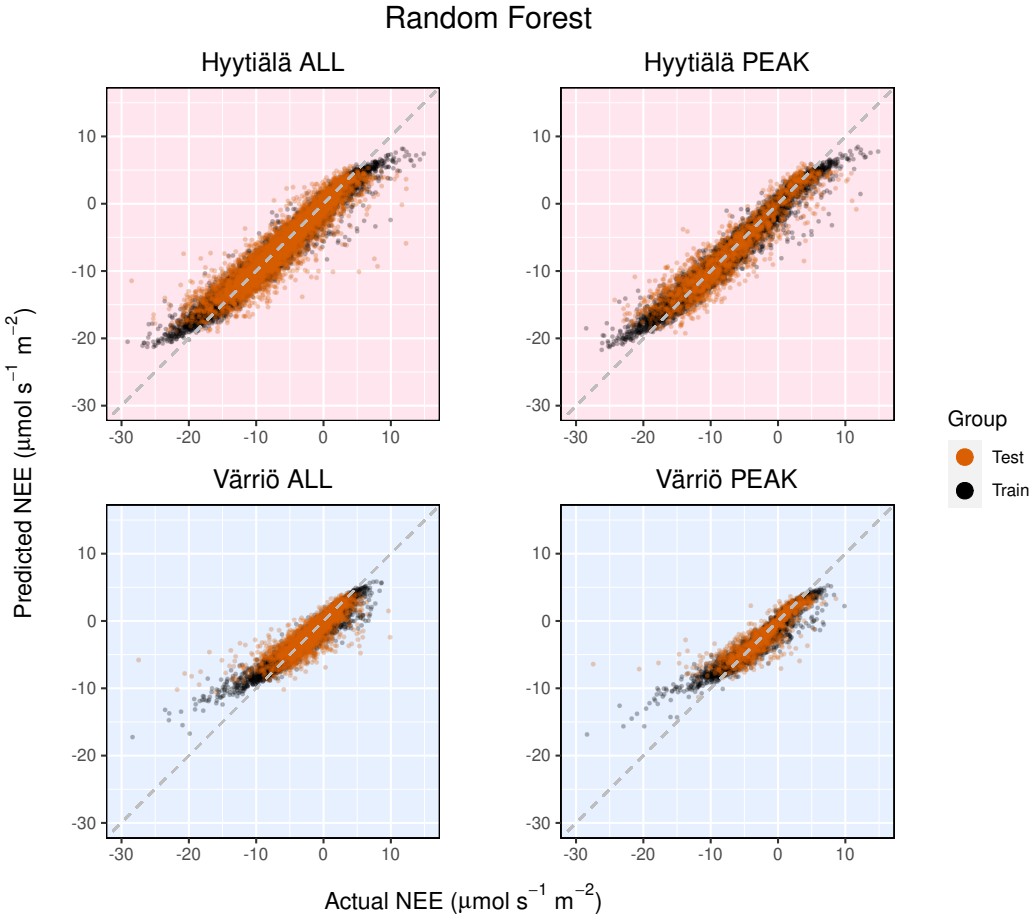

**Figure 3.** Modelled vs measured NEE for Hyytiälä and Värriö on the example of Random Forest. Black points indicate data sets used for model training.





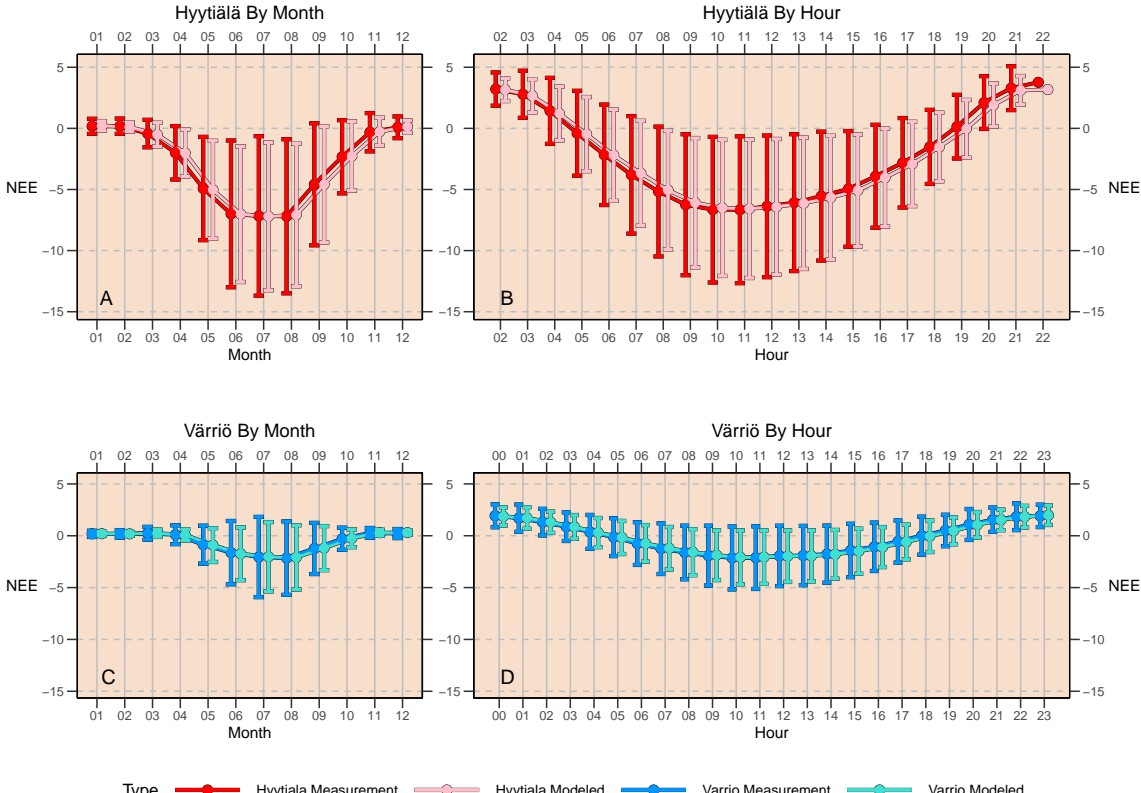

**Figure 4.** Mean diurnal cycle and monthly scatter of daytime NEE as reproduced by RF.





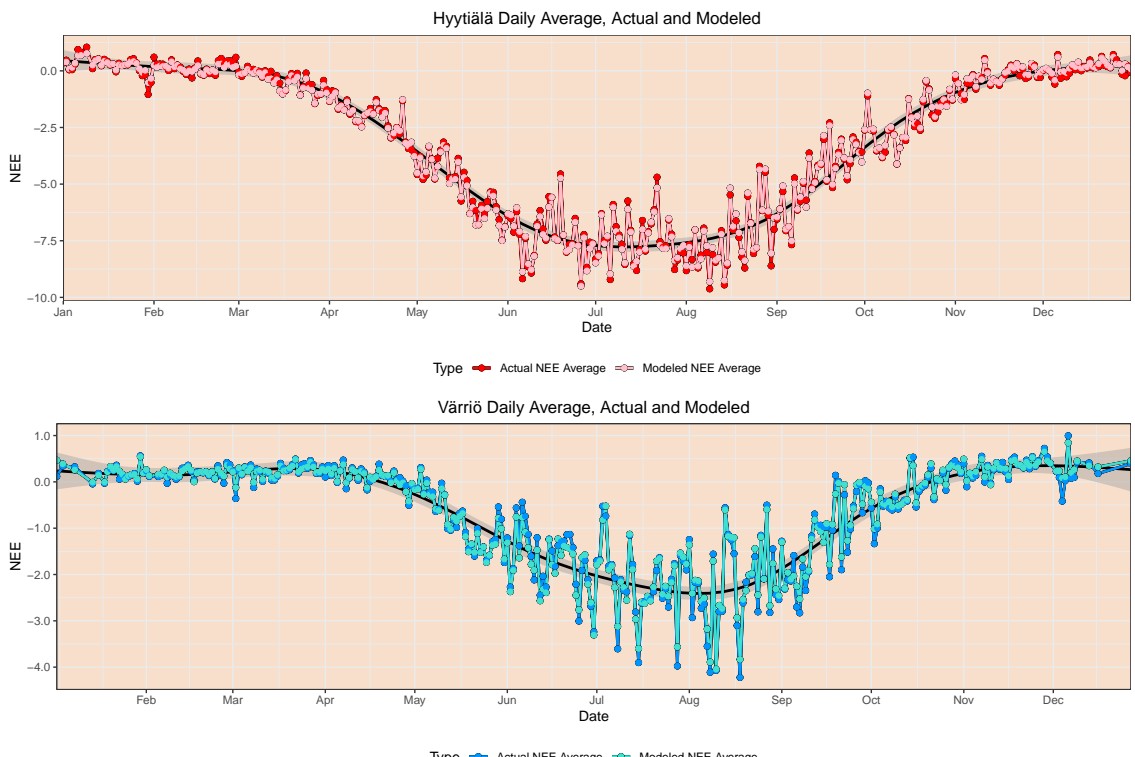

**Figure 5.** Mean annual cycle of daily NEE as reproduced by RF.

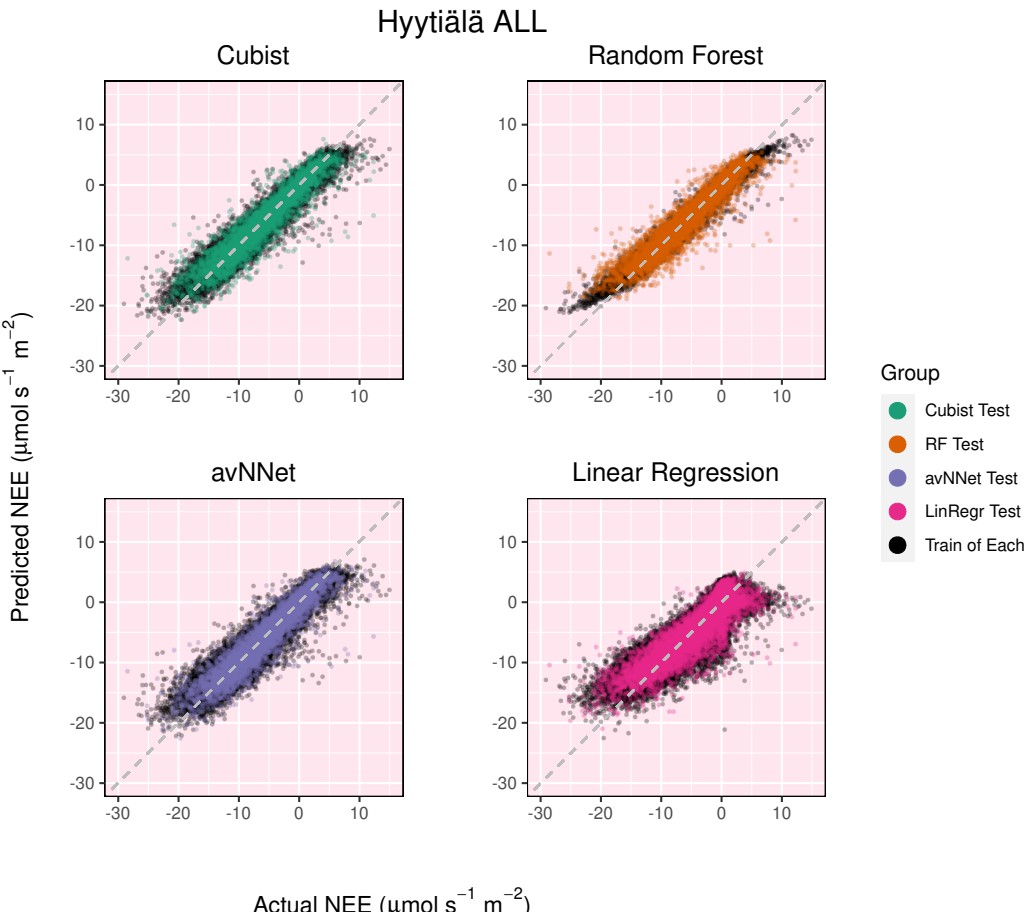

**Figure 6.** Modelled vs measured NEE for Hyytiälä, whole year data set, and all four models. Black points indicate the data sets used for model training.





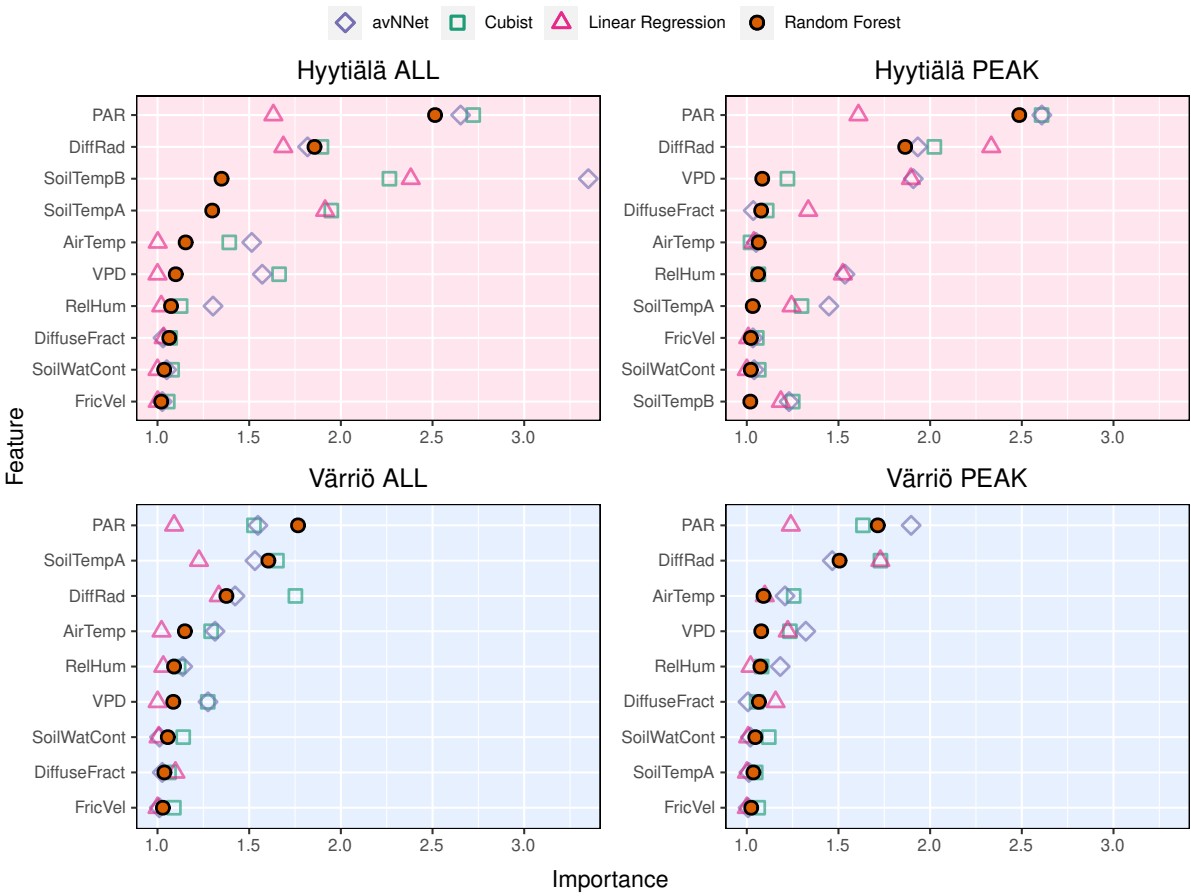

**Figure 7.** Feature importance for all cases and all models. The order of features is in accordance with RF model.





**Figure 8.** ALE for all models, peak growing season in Hyytiälä and Värriö.

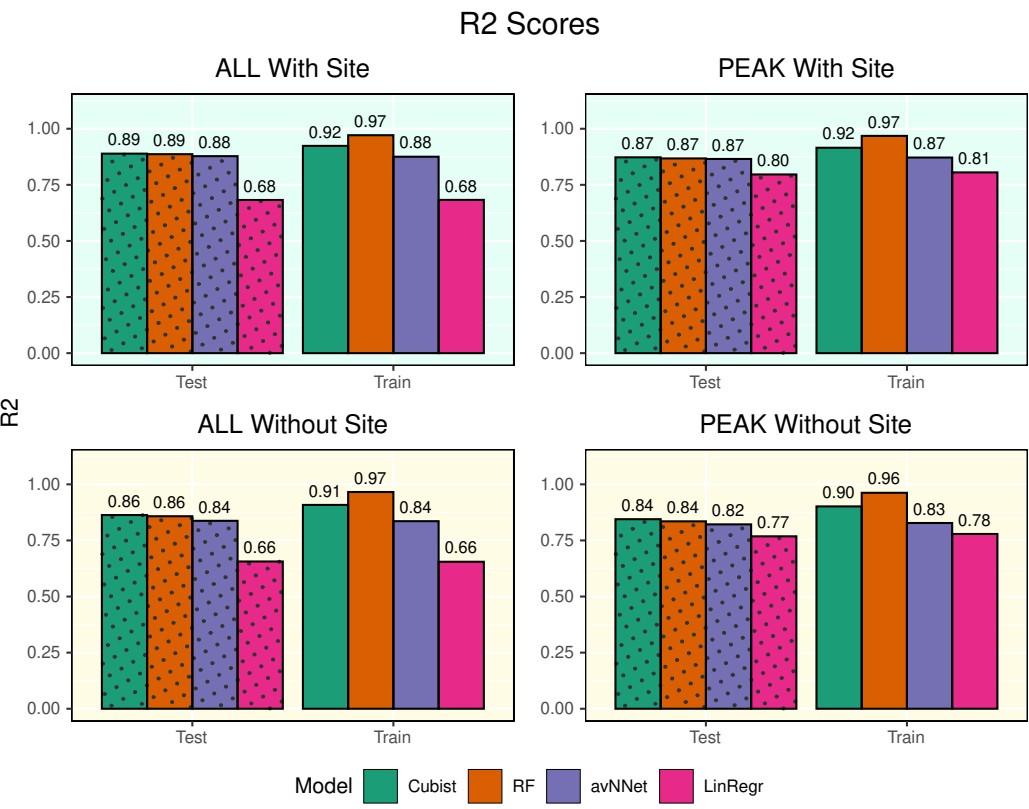

**Figure 9.** $R^2$-coefficients for all models and different data sets. In each of the four panels, the results for the training data set are shown on the right (marked 'Train'), and the results for the test data set are shown on the left (dotted bars, marked 'Test'). 'ALL' denotes the scores for the models using the whole year data sets; 'PEAK' - for the models using the peak growing season data sets. 'With Site' - the input variables contain the information about site, 'Without Site' - no information about site.



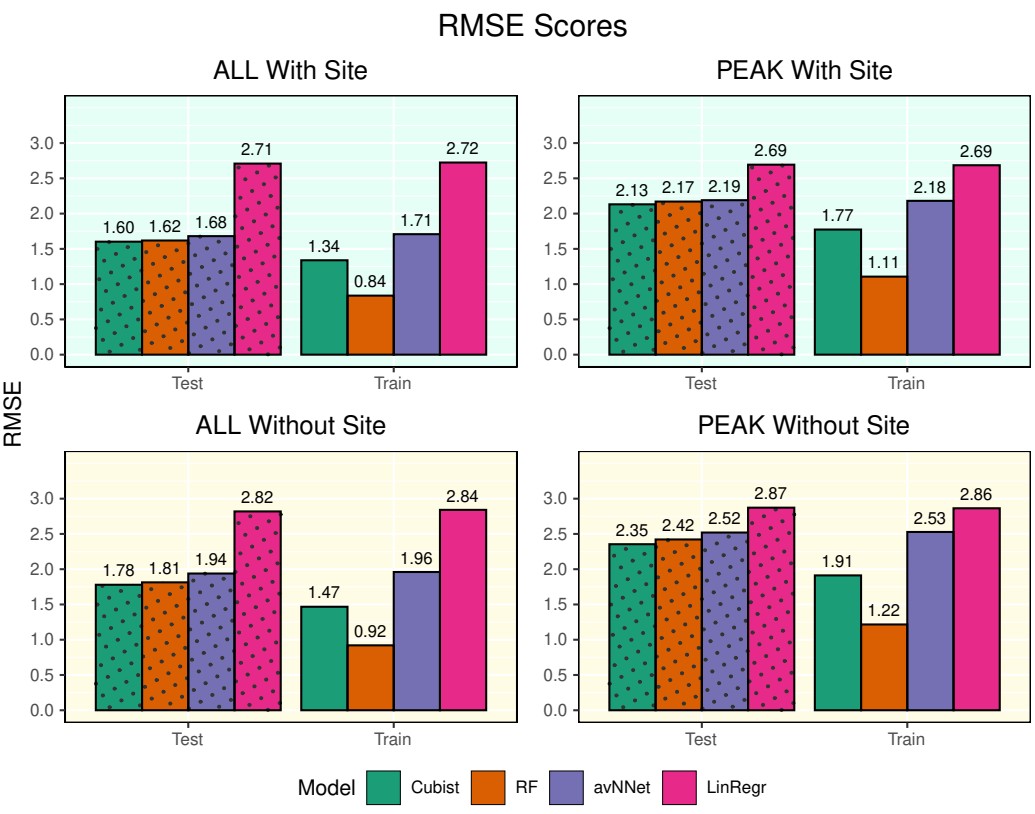

**Figure 10.** RMSE for all models and different data sets. See caption to Fig. 1 for further description.





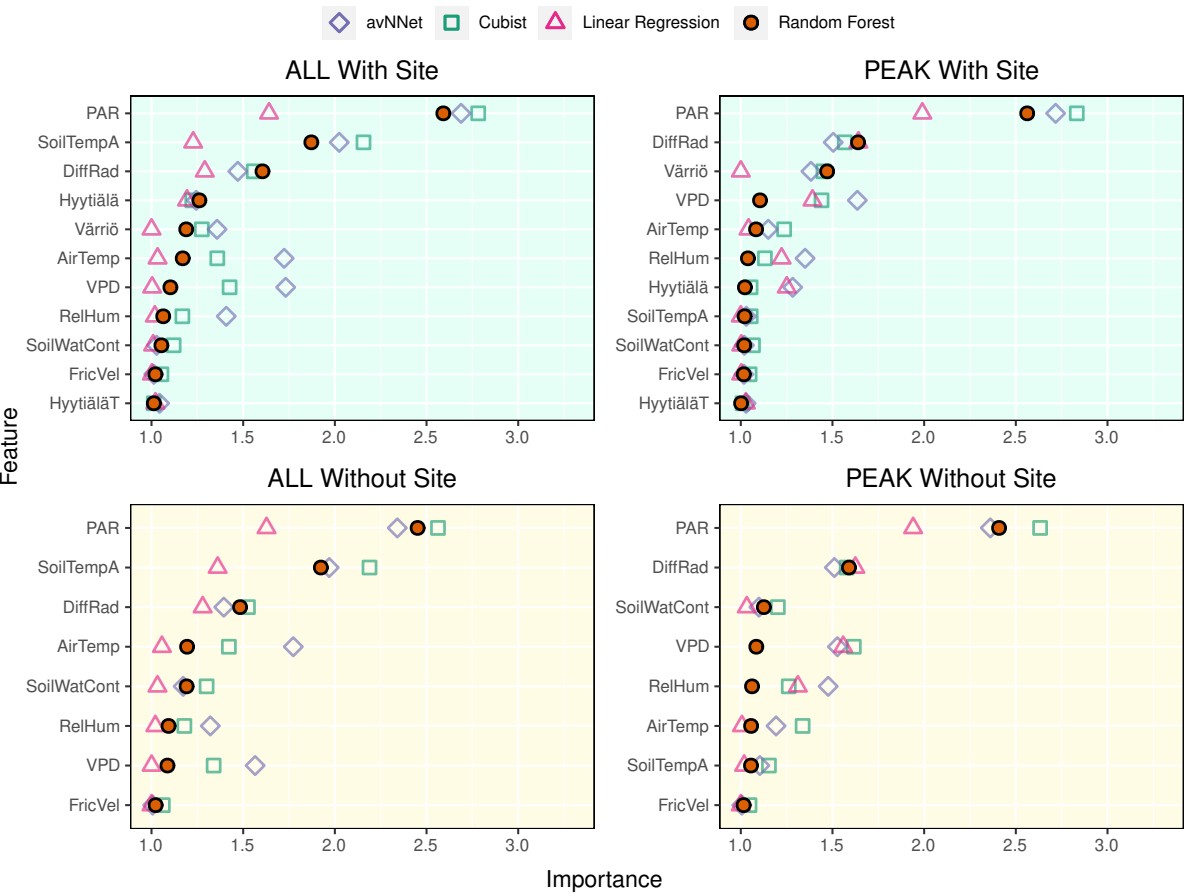

**Figure 11.** Feature importance for site/no-site variables cases. The order of features is in accordance with RF model.





**Figure 12.** ALE for all models, peak growing season, mixed data set with and without site variables.



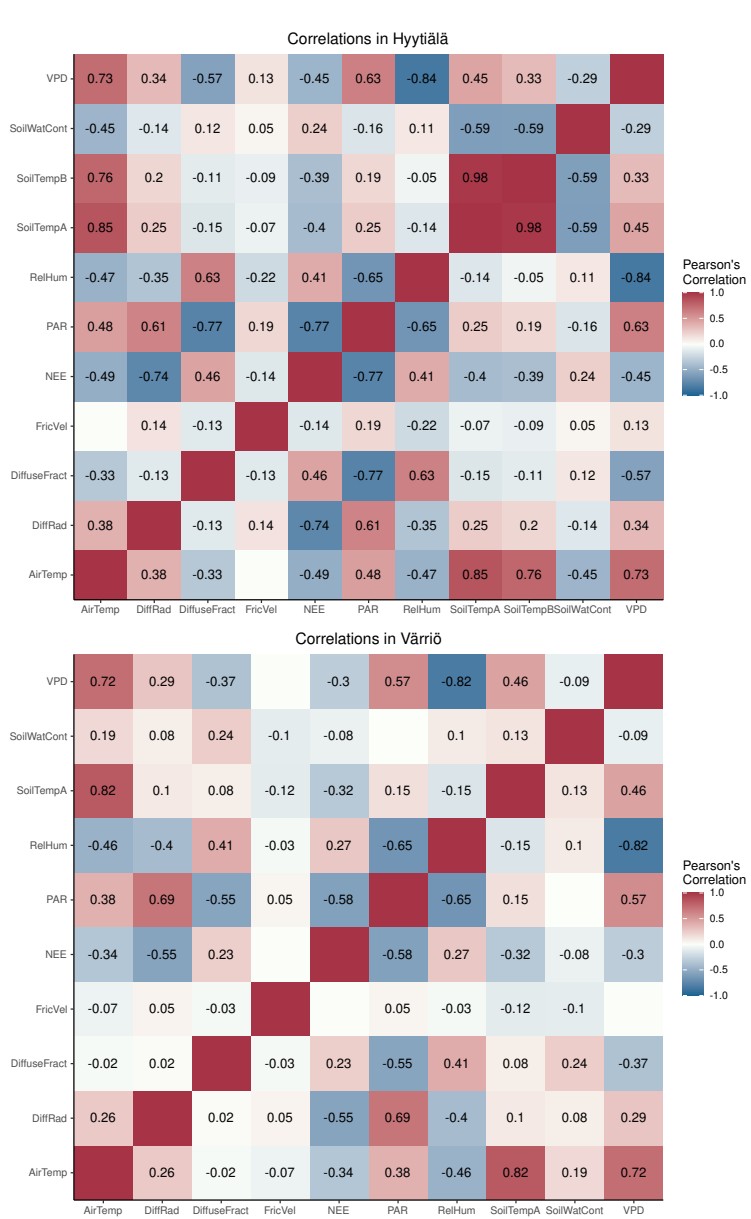

**Figure A1.** Linear correlation between different input variables in Hyytiälä and Värriö.





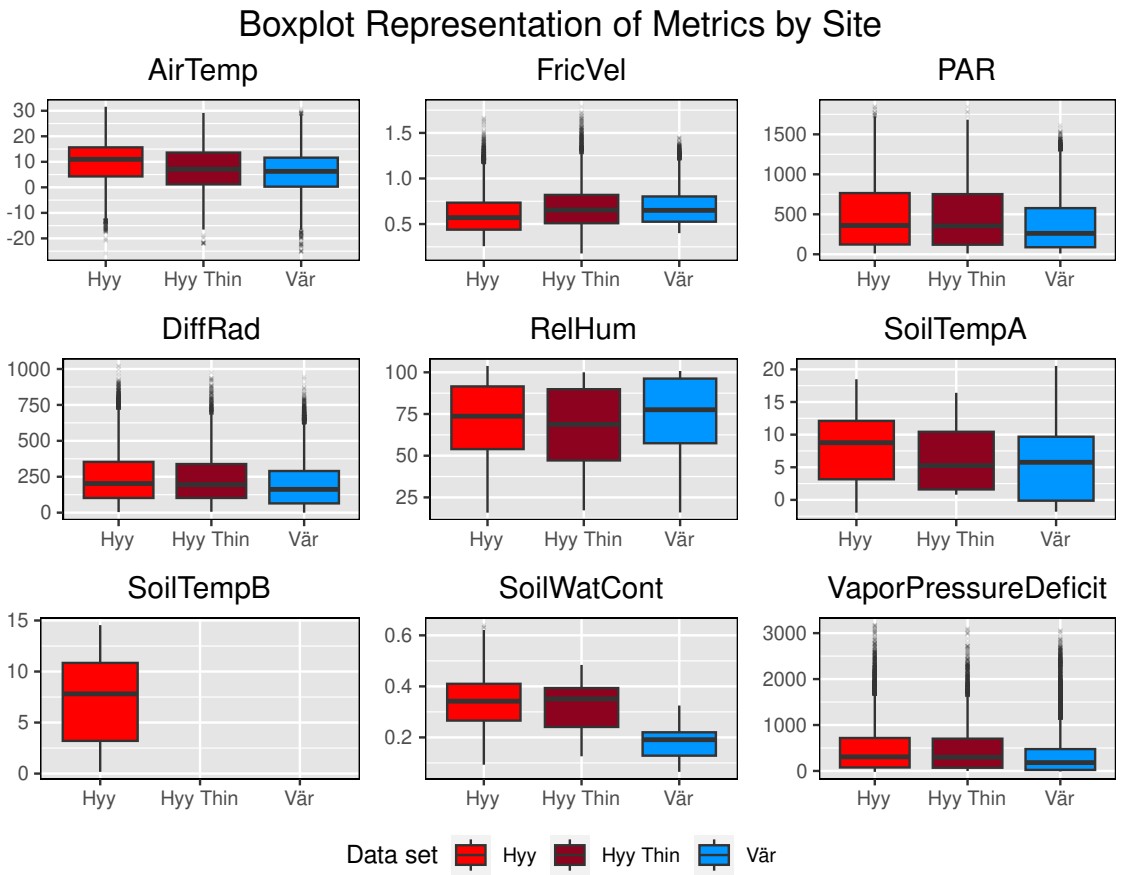

**Figure A2.** Box plots of various input variables comparing Hyytiälä and Värriö.





**Figure A3.** ALE for all models, all-season data set, in Hyytiälä and Värriö.



**Figure A4.** ALE for all models, all season, mixed data set with and without site variables.



**Table A1.** Three most important features for different models and all setups

| Model | RF | Cubist | AvNNet | LinRegr | RF | Cubist | AvNNet | LinRegr |
|---|---|---|---|---|---|---|---|---|
| Peak | Hyytiälä | | | | Värriö | | | |
| P1 | PAR | PAR | PAR | DiffR | PAR | DiffR | PAR | DiffR |
| P2 | DiffR | DiffR | DiffR | VPD | DiffR | PAR | DiffR | PAR |
| P3 | VPD | SoilT$_A$ | VPD | PAR | AirT | AirT | VPD | VPD |
| All | Hyytiälä | | | | Värriö | | | |
| P1 | PAR | PAR | SoilT$_B$ | SoilT$_B$ | PAR | DiffR | PAR | DiffR |
| P2 | DiffR | SoilT$_B$ | PAR | SoilT$_A$ | SoilT$_A$ | SoilT$_A$ | SoilT$_A$ | SoilT$_A$ |
| P3 | SoilT$_B$ | DiffR | DiffR | DiffR | DiffR | PAR | DiffR | DiffR |