# Peer review of "Explainable machine learning for modelling of net ecosystem exchange in boreal forest"

_EGUsphere, 2023_

## Author Comment (AC1)

**Reply to the Referee's comments**

We are grateful to the Referee for reviewing the manuscript and for providing valuable insights, which helped to improve the clarity of the manuscript. Please find below the replies to specific comments:

Introduction: The introduction provides sufficient background information to understand the aim of the paper, however, it often comes across as overly explanatory and some sections could be synthesized and written more concisely to cut down on the word count. Consider revising this section to reduce unnecessary information and improve the flow of the background towards the objectives of the study.

We agree with the comment and have now made the information flow better. Redundant information was removed, and we have focused the text on our research goals.

Line 19-23: Awkward wording. The sentence could flow better, and currently does not come to a satisfying conclusion.

We agree with the comment. We have rearranged the sentence as follows: *The dynamics of the forest carbon cycle and its interaction with various climatic drivers are generally well-understood; however, the complex responses of forests to climate change and their potential to mitigate its impacts keep boreal forests at the forefront of multidisciplinary research. This ongoing interest spans from observational studies to global modeling efforts.*

Line 26: The word "variable" from variable values can be removed, it flows better without and feels like it is already implied from how the sentence is written.

We agree with the comment and have removed this word from the sentence.

Line 49-50: This sentence is jarring and does not give any obvious rationale for why you are now mentioning measurement of different temporal scales. I would consider revising to improve the flow of the paragraph and maybe add a rationale as to why this matters for the paper. Maybe start by discussing the difference between gapfilling and upscaling studies/questions and then discuss how they are typically measured at different temporal scales, so if a researcher would want to look at both they would need multiple scales of data.

We have removed the discussion about the temporal scales from the introduction, because, as the reviewer pointed out, this is not the focus of our manuscript.

Line 74: Referring to subhourly time resolution – Does this relate to the findings of the paper, since you state earlier that upscaling studies typically use longer timescales and upscaling is the part you had the hardest time modeling? Why did you not try using data from multiple temporal resolutions to model both the gapfilling and upscaling if you knew that different resolutions were better for modeling different kinds of data?

We have removed the discussion about temporal scales from the Introduction.

**Line 103–105: It is recommended to be consistent with the ordering of the sites throughout the methods. If you start with SMEAR I (Värriö) then SMEAR II (Hyytiälä), it would be best to always refer to them in that order to avoid confusion.**

We agree with the comment and rearranged the text for the ordering to be more consistent, i.e. Hyytiälä is discussed first and Värriö after it in all notations.

**Line 142: Referring to training vs test data – You should mention what the test sets were as well and how they were selected. It seems like it was 4% of observations used for testing, except for post-thinning Hyytiälä which used 3%, why?**

We understand the confusion, and added clarifications on how much of the data was used for the test/training data: 75% for training and 25% for test for Hyytiälä and Värriö. In the case of the mixed model, 80% of the data was used for training and 20% for testing. These are standard portions used in ML modeling.

**Line 143: Referencing the phrase "individual sites" – Does this separate pre- and post-thinning Hyytiälä, so there are three sets of all season data, and three sets of peak season data, then one set of all data combined, correct? It may be good to be more explicit about what constitutes as individual sites since pre and post-thinning Hyytiälä are from the same site.**

We apologize for not being clear. We added a table that summarizes information about all numerical experiments. We hope that the table will help to distinguish between different research cases. The table is as follows:

**Table 3.** Overview of the training configurations for ML models across different datasets.

| Set | Site/Data Period | Description |
|---|---|---|
| Set 1 | Hyytiälä All | Models trained on the data from pre-thinned Hyytiälä, entire years |
| | Hyytiälä Peak | Models trained on the data from pre-thinned Hyytiälä, peak growing seasons |
| | Värriö All | Models trained on the data from Värriö, entire years |
| | Värriö Peak | Models trained on the data from Värriö, peak growing seasons |
| Set 2 | All Site All | Models trained on the mixed data set from both sites, including post-thinned Hyytiälä, entire years, no site labels |
| | All Site All (Label) | Models trained on the mixed dataset from both sites, including post-thinned Hyytiälä, entire years, sites labels included |
| | All Site Peak | Models trained on the mixed dataset from both sites, including post-thinned Hyytiälä, peak growing seasons, no site labels |
| | All Site Peak (Label) | Models trained on the mixed dataset from both sites, including post-thinned Hyytiälä, peak growing seasons, site labels included |

Line 210: Another small comment, start with the ALE plot paragraph since the method is mentioned first, or mention Permutation Feature Importance first in the preceeding paragraph. It is helpful to be consistent with the order things are discussed.

We agree with this comment and have fixed the ordering to be more consistent regarding the order in which the concepts are discussed, with Feature Importance first, and ALE plots second.

Line 217: Estimates

We included this missing word in the text.

Line 273: Refer to either $R^2$, R-squared, or R-scores throughout the document, do not switch between them.

We thank the referee for the comment. This is fixed to be more consistent across the text, now being referred to exclusively as $R^2$-coefficient.

Line 283: remove ":", instead use ";". Also "accounted here", should be "accounted for here".

We thank the referee for the observation and fixed these typos.

Line 365: I would change the formatting used for this section.

We agree with this note. Formatting was changed here to be consistent with the other parts of the paper.

Line 418–421: I think this distinction is unnecessary, you could just state that you distinguished between the sites by coding them to three dummy variables.

We agree with this comment and have now changed the text as follows: *we introduce three binary variables that identify the site. Three binary variables were used instead of a single categorical one due to some models requiring real numbers as input.*

Line 427: Have you tried running the models after standardizing the number of observations included from each site? Ideally twice to compare the same time periods for Värriö and pre-thinning Hyytiälä, and Värriö and post-thinning Hyytiälä. This could help prevent the scores from following a single site just because it was sampled more.

We thank the referee for this important question. We ran experiments using as a training dataset a balanced dataset with an equal number of data points from Hyytiälä, Värriö and post-thinning Hyytiälä. The results were consistent with the results obtained from a non-balanced dataset, still having $R^2$-coefficient close to that of the site with the better score, Hyytiälä, even though the number decreased marginally (by about 0.02 for non-linear models). RMSE calculated using the unbalanced data set (1.60-1.68, all season) are lower than RMSE calculated using only Hyytiälä data (1.74-1.79), possibly because RMSE for Värriö and Hyytiälä post-thinned data sets are smaller. For the balanced data set, RMSE increases (1.67-1.73) but still below

RMSE for Hyytiälä alone. We therefore conclude that adding more data points from other sites does not necessarily make the predictions worse, especially if there is a site identifier, but makes the predictions of the sites that have additional data points somewhat more accurate. We have added a brief discussion about the balanced data set in Results and score figures in the Supplementary material.

**Line 466–468**: Generally, this should either be in brackets inside the other sentence, or have the brackets removed.

We thank the referee for this observation, the brackets have been removed.

**Figure captions: I believe figures should stand on their own without requiring the reader to have read either the main text or other figure captions. Most of your figure captions are a single line and not very descriptive. Even, as in figure 2, where you ask the reader to refer to an earlier caption is missing from the other figure captions. It would be best if you wrote full captions for all figures.**

We agree with this comment; we have made the figure captions more descriptive.

We thank again the referee for the useful suggestions. We hope that our answers address all the comments raised.

---

## Author Comment (AC2)

**Reply to the Reviewer 2 comments**

We are grateful to the Reviewer for providing valuable insights, which helped to improve the clarity of the manuscript. Please find below our replies to specific comments:

***The abstracts need to be improved by adding more details. The current version is a little vague. For example, the current version only mentioned four machine learning models. It is better to explicitly elaborate what kinds of four machine learning algorithms were used in the manuscript. Furthermore, model performance and statistics are also better to be presented in the abstract. In addition, the manuscript highlights the explainable machine learning to quantify NEE drivers. However, no details on which drivers are most important for NEE predictions in the abstract.***

We thank the reviewer for the comment. We added the requested information in the abstract, which now reads:

'We apply four machine learning models (Cubist, Random Forest, artificial neural network and linear) to predict the NEE of boreal forest ecosystems based on climatic and site variables. We use data sets from two stations in the Finnish boreal forest (southern site Hyytiälä and northern Värriö) and model NEE during the peak growing season and the whole year. All nonlinear models demonstrated similar results with $R^2$=0.88-0.90 for Hyytiälä, and $R^2$=0.70-0.76 for Värriö. Using Explainable Artificial Intelligence methods, we show that three most important input variables during the peak season are photosynthetically active radiation, diffuse radiation and vapour pressure deficit (or air temperature), whereas on the whole year scale, vapour pressure deficit is replaced by soil temperature'.

We are grateful to the reviewer for pointing out the source of uncertainties and potential overfitting, which showed that nonlinear ML models give comparable results, and we changed the text accordingly.

***The manuscript was only conducted once for training and testing dataset splitting. As we know, there are always uncertainties in the data splitting. It is better to conduct data splitting multiple times to also present uncertainties of R2 and RMSE in Figures 1 and 2.***

We thank the reviewer for the valuable suggestion. We agree that uncertainties in data splitting can affect the reliability of our results, and therefore, as was mentioned in Methods, we used k-fold cross-validation in our study. This approach improves the robustness of our findings and allows us to obtain more reliable estimates of $R^2$ and RMSE. However, we have now done model training with multiple different data splits, and observed that they are consistent with our results in Figures 1 and 2 (Figure A). We added uncertainties in Fig. 1-2.

[Figure]

Figure A: ML models trained on different splits of the data. The most variance seen in the scores is in Värriö Peak, which is most likely due to it being the smallest data set. Overall, the scores are consistent with the results shown in the manuscript.

***From Figures 1 and 2, it seems that some machine learning models are overfitting, which means that training performance is much better than testing performance. It is better to tune hyperparameters of machine learning models to avoid machine learning model overfitting.***

We thank the reviewer for this comment. We have used k-fold cross-validation, as well as internal model hyperparameter tuning, which both reduce the likelihood of overfitting [1,2]. We observed the mean 0.16 difference in $R^2$ score between the testing and training performance of Värriö setup for Random Forest and Cubist. Model hyperparameter tuning resulted in poorer training performance would always result in poorer test performance.

Figure B below illustrates the Random Forest performance on Värriö All setup when different hyperparameters that can have effect on overfitting are tuned. The most crucial hyperparameter in regards of overfitting, min.node.size [2], shows that increasing it decreases both testing and training model performance, as well as the difference between them. This points at decreasing overfitting. At the largest value of min.node.size used here, we got the model test $R^2$ score of 0.70, i.e., closer to the neural network result, which was the lowest among the nonlinear models.

[Figure]

Figure B: Experimental tuning of Random Forest hyperparameters. Tuning these hyperparameters can reduce overfitting, but in this case reducing the overfitting resulted in poorer training and test scores, indicating overall poorer fit while the difference between training and test scores did become smaller.

***Figures 3 and 6. There are too many points in the scatter plots. It is better to use density scatter plots to illustrate the results****.*

We agree with the notion that there are too many points in the scatter plots to the degree that loading the images takes too long. However, after comparison of scatter and density plots, we felt that scatter plots better tell the story of how the different datasets deviate from each other than density plots. We have reduced the file size instead to make inspecting the images smoother.

***Figures 4 and 5. It is better to add units to the y-axis***.

We added the units to the y-axis in both images.

***Figures 7 and 11. Better to add uncertainty bars, once you have done different data splitting***.

We thank the reviewer for the suggestion. While we recognize the importance of representing variability with uncertainty bars, our study uses a consistent data split across all XAI methods to ensure comparability. This approach helps us to maintain control and attribute differences directly to the XAI techniques rather than data variability. However, as the $R^2$ and RMSE scores have some deviation for Värriö PEAK data set, we decided to add make feature importance for various different splits to ensure that the results are consistent with different data splits. The results can be seen in Figure C:

[Figure]

Figure C: Feature importance for Värriö ALL and Värriö PEAK with uncertainty bars, where Feature Imporance was done to multiple different data splits of these data sets. All were done on test data, data that the model was not used to train on. The points display the mean value, while the error bars display the maximum and minimum values. Here the ordering is based on the mean importance acrossa all models, instead of just Random Forests, as is in the manuscript. The results are consistent with what we present in the manuscript.

***Figure A1. It is better to add the significance levels for the correlation analysis.***

We thank the reviewer for this suggestion. Given that almost all the correlation coefficients in our analysis show p-values significantly lower than the conventional threshold (e.g., $p < 0.05$), indicating statistical significance, we added asterisks in Figure A1 where needed to mark nonsignificant values.

We thank the reviewer again for the useful suggestions. We hope that our answers address all the comments raised.

**References**

[1] Bengio Y, Grandvalet Y. No Unbiased Estimator of the Variance of K-Fold Cross-Validation. Journal of Machine Learning Research 5 (2004) 1089–1105.

[2] Barreñada, L., Dhiman, P., Timmerman, D., Boulesteix, A. L., & Van Calster, B. (2024). Understanding random forests and overfitting: a visualization and simulation study. *arXiv preprint arXiv:2402.18612*.

---

## Author Response (AR1)

**Answer to the Editor's comments**

We are grateful to the Editor for raising the points that require some further discussion.

**Please note, the rating of your manuscript is not overly high and this maybe due to weaknesses in the presentation that you now opt for improving.**

It seems that the overall prevailing grade that the reviewers gave for our manuscript was 'good', but the second reviewer rated the 'significance of the topic' to be only 'fair' although this reviewer also considered the study 'interesting'. We hope the revised manuscript will succeed better in showing the topic's significance. In our study, we apply several popular machine learning models to model net ecosystem exchange in boreal forests. These machine learning models are often used as 'black box' models, meaning that their decision making is often not well understood. The novelty of our manuscript is that we visualize and examine the models' decisions on different input parameters based on the existing knowledge about ecosystem functioning. The study is truly multidisciplinary as the results produced by computer scientists are interpreted by atmospheric and forest scientists. We hope that the revised version of our manuscript accounting for the Reviewers' and Editor's comments will improve the low rating and make our study acceptable for publication.

**But you may consider once more, whether you can discuss and summarise the most important achievements of the study a bit clearer. For example when discussing the importance values (Fig. 7 and Fig. C), add explanation, whether / why these findings make sense (and explainable ML a good choice). Did you learn anything unexpected form the explanations from ML?**

We thoroughly went through the text and modified it following the Editor's suggestions. The discussion, addressing point by point the dependence of NEE on all input parameters for separate data sets, related to Figs. 7 and 8, was provided in subsection 3.1.2, and for the mixed data sets – in subsection 3.2.2 of the previous manuscript version. To improve the clarity of the discussions, we have split them into separate subsections focusing on feature importance (current subsections 3.1.2, 3.2.2) and ALE (current subsections 3.1.3, 3.2.3). We have also introduced similar logic in the subsections addressing ALE, starting the discussions with the most important variables as suggested by the feature importance diagrams.

In the case of separate sites and seasons, all the nonlinear models choose similar most powerful explanatory parameters (Lines 311-327), and compared to existing knowledge on ecosystem functioning, there were largely no surprises, which means that models in general work well. The choice of important variables becomes even more aligned when the mixed data sets are used (Lines 465-478). More surprising was to see how some models treat interdependent parameters (e.g., soil temperatures at different depths or VPD/RH), as they may produce strong but opposite dependencies of NEE on these parameters (Lines 373-387, 502-504). Another interesting finding was that for some models, soil water content that was low in importance among the set of features for separate sites suddenly becomes one of the most important variables for mixed data sets (Lines 471-473). This is accompanied by a drastic change of ALE for this variable (Lines 495-501) in all the models. As soil water content is the only variable that has a clear difference between Hyytiälä and Värriö data sets (Fig. A2), we interpret this new high position of soil water content in the

feature importance diagrams so that the models treat it as a site parameter to distinguish between the Hyytiälä and Värriö data sets. This interpretation is supported by soil water content losing importance when site parameters are added as input variables.

We have also included text related to the added error bars due to various data splits (Lines 332-336): 'For Värriö, Cubist and avNNet place interdependent VPD, RH and air temperature in the feature importance diagram within the error bar from each other. Relatively large error bars for these variables suggest that the models seem to have difficulties ranking them, as their order may likely change depending on the data split. At the same time, the error bars are smallest for Random Forest, which seems to be more confident than the other nonlinear models in its treatment of interdependent variables.'

**For example you find that the intensity of diffuse radiation has a higher importance than the fraction of diffuse radiation. Why that?**

We added the following discussion on this in the manuscript:

Lines 337-342: Suppose the model chooses one variable before another correlated one. In that case, the second one can be placed low in the feature importance diagram, as the model, in principle, does not need it anymore. This does not mean, however, that one of the correlated variables explains NEE clearly better than the other: for example, Moffat et al. (2010) showed, using an artificial neural network, that intercorrelated diffuse fraction and diffuse radiation (as well as intercorrelated VPD and RH) have the same explanatory power for the summertime forest NEE, and can be used interchangeably. However, all our models place diffuse PAR higher than diffuse fraction, and they typically place VPD higher than RH.

Lines 391-394: Gross primary production in Hyytiälä has its minimum at the low diffuse PAR and a maximum at the high diffuse PAR compared to the weak parabolic dependence on diffuse fraction (Ezhova et al., 2018; Neimane-Šroma et al., 2024). That may be why the models choose diffuse PAR over diffuse fraction. Most models could then deem the diffuse fraction relatively unimportant as they already use diffuse PAR.

**Then you explain that the VPD effect is rather explained by temperature than by relative humidity. Why that?**

VPD is a function of both relative humidity and temperature as follows from eq. (2), and it is strongly correlated with both (Fig. A1). VPD influences photosynthesis via stomatal control. RH contribution to NEE is basically via this VPD effect on photosynthesis, whereas temperature, in addition, affects respiration.

We mention that VPD is dependent on both variables in various parts of the manuscript:

Lines 322-323: 'It is good to note that VPD is calculated based on air temperature (see Sec. 2.1), so these variables are not independent.'

Lines 332-333: 'For Värriö, Cubist and avNNet place interdependent VPD, RH and air temperature in the feature importance diagram within the error bar from each other'.

Lines 395-397: 'RH directly influences VPD through a linear relationship (eq. (2), Fig. A1). The higher the RH, the closer ambient air is to saturation, and VPD, in this case, is small. Low RH, vice versa, favors higher VPD values'.

**Finally, you include the friction velocity as a variable, which doesn't yield a high importance score. This might be even trivial as the u\* filtering is applied to the data sets, i.e. to exactly remove any relationship between u\* and NEE.**

We chose u\* as one of the parameters following the setup in Moffat et al. (2010) to be able to compare with this study (and got the similar result that the variable is unimportant). While it is true that filtering is applied to exclude the lowest u\* corresponding to non-turbulent conditions from the data sets, some relationship might still be there for higher values: actually, for Hyytiälä, there is some weak positive correlation, which we briefly mention in lines 405-409. The conclusion about NEE not depending on u\* may serve as an additional checkpoint for the quality of the data set.

**In your reply to Reviewer 2 you mention that fitting hyper-parameters reduced overall performance, which is to be expected. But you do not seem to explain, why and for which application a result less prone to over-fitting might be a better choice.**

We are grateful to the Editor for stimulating the discussion on overfitting.

The K-fold cross-validation technique, which we have now used to find hyperparameters, can also be used to estimate the model performance on an unknown data set. K-fold cross-validation method shuffles the data set randomly and splits it into K groups or folds. First, each fold is taken as a holdout, while the model is fit on the rest of the folds, and then the model is evaluated on the holdout set. This procedure is repeated R times. Each time, we can calculate R2-scores and RMSE corresponding to the evaluation (holdout) data set. The average accuracy metrics obtained in such a procedure provide a reliable estimate of how the model is expected to perform on an unknown or test data set (e.g., Refailzadeh et al., 2009).

The R2-scores obtained from the cross-validation of our models' performance on the data sets from separate stations are now reported in Fig. 1 of this document. For all the models, the metrics agree with the scores reported in the manuscript for the test data set. The estimate obtained from cross-validation is even slightly lower than the results obtained on the test data set, likely due to the smaller size of the holdout subsets from the folding procedure. Therefore, we can conclude that all the models perform on the test data set with their expected scores.

However, when a final trained model is applied again to the training data set, the resulting scores sometimes can be high, as we see for both models based on regression trees, suggesting some overfitting. Nevertheless, it is not obvious if this degree of overfitting is necessarily bad, as mentioned by e.g., Zhang et al., 2023: 'Note that overfitting is not always a bad thing. In deep learning especially, the best predictive models often perform far better on training data than on holdout data'. Based on the abovementioned arguments, we prefer keeping the hyperparameters in our models unchanged.

[Figure]

Fig. 1. R2-scores obtained by different ML models (see the legend) on test data (colored bars) and from the cross-validation procedure on the training data set (dotted bars). Panels correspond to the following setups: Hyytiälä ALL - the whole-year Hyytiälä data set, Hyytiälä PEAK – the peak season Hyytiälä data set, Värriö ALL – the whole-year Värriö data set, Värriö PEAK – the peak season Värriö data set.

Related to the editor's comment 'for which application a result less prone to over-fitting might be a better choice', there is unfortunately no general solution.

We added the following discussion in the manuscript: 'The difference in scores between the training and test data sets is called generalization error. In some cases, large generalization error points to overfitting, i.e., the model learns the training data set too well and then performs poorly on the test data set. To tune hyperparameters and estimate expected model performance, we applied K-fold cross-validation, see subsection 2.3. Additionally, we tried different splits of the data into training and test data sets, which showed that the variation of the resulting $R^2$-coefficients and RMSE was small. Finally, we obtained similar accuracy metrics on the test data sets from different nonlinear ML models, suggesting that our results are robust.'

**Please reconsider the Reviewer 2's comment on the advantages of density contour plots over scatter plots. I do not think that the main point was the file size, but the loss of information when too many points are covered and thus hidden by the points on top in scatter plots. Here the density contour plot gives more information than scatter plots. Your explanation on why the scatter plots are better is a bit vague. Please provide the two alternatives and then justify your choice.**

We thank the reviewer and the editor for the comment. Below, we have provided a density contour plot in Figure A and a density scatter plot in Figure B to illustrate what the alternatives would look like compared to the plot we considered using, Figure C.

Overall, the problem of showing two scatter plots in one figure is hard to solve in this context as the distributions of points are very close to each other (Fig. C). The difference is in the noisy outlier points, which disappear entirely when we try contour plots and become almost invisible if we make one set of points transparent.

Our aim with scatter plot Fig. 3 was:
1. to see if the correlations follow 1:1 lines;
2. to illustrate obtained R2 and RMSE results for test and train data sets and all setups. The figure helps to understand that different R2 could simply be the result of different data range for Värriö and Hyytiälä, and it is also helpful when discussing possible overfitting issues.

In Fig. 6, we compare the performance of different models on the same setup. The figure clearly shows that the RF and linear models look a bit different from the other models. In the case of RF, there are fewer black points around orange points, indicating that the RMSE in the training data set is smaller compared to the test data, while there are visible clouds of black points around color points for other models.

Overall, our suggestion is to visualize density distributions of points on the sides of the figures and reduce the size of the figures as shown in Fig. C of the current document.

[Figure]

Figure A: A density contour plot alternative to Figure 3 of the manuscript.

[Figure]

Figure B: A density scatter plot alternative to Figure 3 of manuscript.

[Figure]

Figure C: The scatter plot used in the revised manuscript as Figure 3.

Finally, we have modified the abstract compared to the version sent to Reviewer 2. We have improved the clarity and included a sentence about the mixed data set results, which were previously omitted. In the abstract, we have also updated the R-scores compared to how they were presented in the response to Reviewer 2. In this response document, we had a small error in Figure A illustrating the results for Värriö Peak, with the scores lower (0.71-0.74 while the correct numbers were 0.73-0.76) compared to the original manuscript and to its current revised version.

We thank again the Editor for the useful suggestions. We hope that you will find that the present manuscript addresses all the comments raised.

**References**

Refaeilzadeh, P., Tang, L., Liu, H. (2009). Cross-Validation. In: LIU, L., ÖZSU, M.T. (eds) Encyclopedia of Database Systems. Springer, Boston, MA. https://doi.org/10.1007/978-0-387-39940-9_565

Zhang, A., Lipton, Z. C., Li, M., & Smola, A. J. (2023). *Dive into deep learning.* Cambridge University Press.